# Efficient Sampling for Learning Sparse Additive Models in High Dimensions

**Hemant Tyagi**
ETH Zürich
htyagi@inf.ethz.ch

**Andreas Krause**
ETH Zürich
krausea@ethz.ch

**Bernd Gärtner**
ETH Zürich
gaertner@inf.ethz.ch

## Abstract

We consider the problem of learning sparse additive models, i.e., functions of the form: $f(\mathbf{x}) = \sum_{l \in S} \phi_l(x_l)$, $\mathbf{x} \in \mathbb{R}^d$ from point queries of $f$. Here $S$ is an unknown subset of coordinate variables with $|S| = k \ll d$. Assuming $\phi_l$'s to be smooth, we propose a set of points at which to sample $f$ and an efficient randomized algorithm that recovers a *uniform approximation* to each unknown $\phi_l$. We provide a rigorous theoretical analysis of our scheme along with sample complexity bounds. Our algorithm utilizes recent results from compressive sensing theory along with a novel convex quadratic program for recovering robust uniform approximations to univariate functions, from point queries corrupted with arbitrary bounded noise. Lastly we theoretically analyze the impact of noise – either arbitrary but bounded, or stochastic – on the performance of our algorithm.

## 1 Introduction

Several problems in science and engineering require estimating a real-valued, non-linear (and often non-convex) function $f$ defined on a compact subset of $\mathbb{R}^d$ in high dimensions. This challenge arises, e.g., when characterizing complex engineered or natural (e.g., biological) systems [1, 2, 3]. The numerical solution of such problems involves learning the unknown $f$ from point evaluations $(\mathbf{x}_i, f(x_i))_{i=1}^n$. Unfortunately, if the only assumption on $f$ is of mere smoothness, then the problem is in general intractable. For instance, it is well known [4] that if $f$ is $C^s$-smooth then $n = \Omega((1/\delta)^{d/s})$ samples are needed for uniformly approximating $f$ within error $0 < \delta < 1$. This exponential dependence on $d$ is referred to as the curse of dimensionality.

Fortunately, many functions arising in practice are much better behaved in the sense that they are intrinsically low-dimensional, i.e., depend on only a small subset of the $d$ variables. Estimating such functions has received much attention and has led to a considerable amount of theory along with algorithms that do not suffer from the curse of dimensionality (cf., [5, 6, 7, 8]). Here we focus on the problem of learning one such class of functions, assuming $f$ possesses the *sparse* additive structure:

$$f(x_1, x_2, \ldots, x_d) = \sum_{l \in S} \phi_l(x_l); \quad S \subset \{1, \ldots, d\}, |S| = k \ll d. \tag{1.1}$$

Functions of the form (1.1) are referred to as sparse additive models (SPAMs) and generalize sparse linear models to which they reduce to if each $\phi_l$ is linear. The problem of estimating SPAMs has received considerable attention in the regression setting (cf., [9, 10, 11] and references within) where $(x_i, f(x_i))_{i=1}^n$ are typically i.i.d samples from some unknown probability measure $\mathbb{P}$. This setting, however, does not consider the possibility of sampling $f$ at specifically chosen points, tailored to the additive structure of $f$. In this paper, we propose a strategy for querying $f$, together with an efficient recovery algorithm, with much stronger guarantees than known in the regression setting. In particular, we provide the first results guaranteeing uniformly accurate recovery of *each individual component* $\phi_l$ of the SPAM. This can be crucial in applications where the goal is to not merely approximate $f$, but gain insight into its structure.

**Related work.** SPAMs have been studied extensively in the regression setting, with observations being corrupted with random noise. [9] proposed the COSSO method, which is an extension of the Lasso to the reproducing kernel Hilbert space (RKHS) setting. A similiar extension was considered in [10]. In [12], the authors propose a least squares method regularized with smoothness, with each $\phi_l$ lying in an RKHS and derive error rates for estimating $f$, in the $L^2(\mathbb{P})$ norm[1]. [13, 14] propose methods based on least squares loss regularized with sparsity and smoothness constraints. [13] proves consistency of its method in terms of mean squared risk while [14] derives error rates for estimating $f$ in the empirical $L^2(\mathbb{P}_n)$ norm [1]. [11] considers the setting where each $\phi_l$ lies in an RKHS. They propose a convex program for estimating $f$ and derive error rates for the same, in the $L^2(\mathbb{P}), L^2(\mathbb{P}_n)$ norms. Furthermore they establish the minimax optimality of their method for the $L^2(\mathbb{P})$ norm. For instance, they derive an error rate of $O((k \log d/n) + kn^{-\frac{2s}{2s+1}})$ in the $L^2(\mathbb{P})$ norm for estimating $C^s$ smooth SPAMs. An estimator similar to the one in [11] was also considered by [15]. They derive similar error rates as in [11], albeit under stronger assumptions on $f$.

There is further related work in approximation theory, where it is assumed that $f$ can be sampled at a desired set of points. [5] considers a setting more general than (1.1), with $f$ simply assumed to depend on an unknown subset of $k \ll d$-coordinate variables. They construct a set of sampling points of size $O(c^k \log d)$ for some constant $c > 0$, and present an algorithm that recovers a uniform approximation[2] to $f$. This model is generalized in [8], with $f$ assumed to be of the form $f(x) = g(\mathbf{A}x)$ for unknown $\mathbf{A} \in \mathbb{R}^{k \times d}$; each row of $\mathbf{A}$ is assumed to be sparse. [7] generalizes this, by removing the sparsity assumption on $\mathbf{A}$. While the methods of [5, 8, 7] could be employed for learning SPAMs, their sampling sets will be of size exponential in $k$, and hence sub-optimal. Furthermore, while these methods derive uniform approximations to $f$, they are unable to recover the individual $\phi_l$'s.

**Our contributions.** Our contributions are threefold:

1. We propose an efficient algorithm that queries $f$ at $O(k \log d)$ locations and recovers: (i) the active set $S$ along with (ii) a *uniform approximation* to each $\phi_l$, $l \in S$. In contrast, the existing error bounds in the statistics community [11, 12, 15] are in the much weaker $L^2(\mathbb{P})$ sense. Furthermore, the existing theory in both statistics and approximation theory provides explicit error bounds for recovering $f$ and not the individual $\phi_l$'s.

2. An important component of our algorithm is a novel convex quadratic program for estimating an unknown univariate function from point queries corrupted with *arbitrary* bounded noise. We derive rigorous error bounds for this program in the $L^\infty$ norm that demonstrate the robustness of the solution returned. We also explicitly demonstrate the effect of noise, sampling density and the curvature of the function on the solution returned.

3. We theoretically analyze the impact of additive noise in the point queries on the performance of our algorithm, for two noise models: arbitrary bounded noise and stochastic (iid) noise. In particular for additive Gaussian noise, we show that our algorithm recovers a robust *uniform approximation* to each $\phi_l$ with at most $O(k^3 (\log d)^2)$ point queries of $f$. We also provide simulation results that validate our theoretical findings.

## 2 Problem statement

For any function $g$ we denote its $p^{\text{th}}$ derivative by $g^{(p)}$ when $p$ is large, else we use appropriate number of prime symbols. $\| g \|_{L^\infty[a,b]}$ denotes the $L^\infty$ norm of $g$ in $[a,b]$. For a vector $\mathbf{x}$ we denote its $\ell_q$ norm for $1 \leq q \leq \infty$ by $\| \mathbf{x} \|_q$.

We consider approximating functions $f : \mathbb{R}^d \to \mathbb{R}$ from point queries. In particular, for some unknown active $S \subset \{1, \ldots, d\}$ with $|S| = k \ll d$, we assume $f$ to be of the additive form: $f(x_1, \ldots, x_d) = \sum_{l \in S} \phi_l(x_l)$. Here $\phi_l : \mathbb{R} \to \mathbb{R}$ are the individual univariate components of the model. Our goal is to query $f$ at suitably chosen points in its domain in order to recover an estimate $\phi_{\text{est},l}$ of $\phi_l$ in a compact subset $\Omega \subset \mathbb{R}$ for each $l \in S$. We measure the approximation error in the $L^\infty$ norm. For simplicity, we assume that $\Omega = [-1, 1]$, meaning that we guarantee an upper

bound on: $\| \phi_{\text{est},l} - \phi_l \|_{L^\infty[-1,1]}$ ; $l \in S$. Furthermore, we assume that we can query $f$ from a slight enlargement: $[-(1+r),(1+r)]^d$ of $[-1,1]^d$ for[3] some small $r > 0$. As will be seen later, the enlargement $r$ can be made arbitrarily close to 0. We now list our main assumptions for this problem.

1. Each $\phi_l$ is assumed to be sufficiently smooth. In particular we assume that $\phi_l \in C^5[-(1+r),(1+r)]$ where $C^5$ denotes five times continuous differentiability. Since $[-(1+r),(1+r)]$ is compact, this implies that there exist constants $B_1,\ldots,B_5 \geq 0$ so that

$$\max_{l \in S} \| \phi_l^{(p)} \|_{L^\infty[-(1+r),(1+r)]} \leq B_p; \quad p = 1,\ldots,5. \tag{2.1}$$

2. We assume each $\phi_l$ to be centered in the interval $[-1,1]$, i.e. $\int_{-1}^{1} \phi_l(t)dt = 0$; $l \in S$. Such a condition is necessary for unique identification of $\phi_l$. Otherwise one could simply replace each $\phi_l$ with $\phi_l + a_l$ for $a_l \in \mathbb{R}$ where $\sum_l a_l = 0$ and unique identification will not be possible.

3. We require that for each $\phi_l$, $\exists I_l \subseteq [-1,1]$ with $I_l$ being connected and $\mu(I_l) \geq \delta$ so that $|\phi_l'(x)| \geq D$ ; $\forall x \in I_l$. Here $\mu(I)$ denotes the Lebesgue measure of $I$ and $\delta, D > 0$ are constants assumed to be known to the algorithm. This assumption essentially enables us to detect the active set $S$. If say $\phi_l'$ was zero or close to zero throughout $[-1,1]$ for some $l \in S$, then due to Assumption 2 this would imply that $\phi_l$ is zero or close to zero.

We remark that it suffices to use estimates for our problem parameters instead of exact values. In particular we can use upper bounds for: $k$, $B_p$; $p = 1,\ldots,5$ and lower bounds for the parameters: $D, \delta$. Our methods and results stated in the coming sections will remain unchanged.

## 3 Our sampling scheme and algorithm

In this section, we first motivate and describe our sampling scheme for querying $f$. We then outline our algorithm and explain the intuition behind its different stages. Consider the Taylor expansion of $f$ at any point $\xi \in \mathbb{R}^d$ along the direction $\mathbf{v} \in \mathbb{R}^d$ with *step size*: $\epsilon > 0$. For any $C^p$ smooth $f$; $p \geq 2$, we obtain for $\zeta = \xi + \theta\mathbf{v}$ for some $0 < \theta < \epsilon$ the following expression:

$$\frac{f(\xi + \epsilon\mathbf{v}) - f(\xi)}{\epsilon} = \langle \mathbf{v}, \bigtriangledown f(\xi) \rangle + \frac{1}{2}\epsilon\mathbf{v}^T \bigtriangledown^2 f(\zeta)\mathbf{v}. \tag{3.1}$$

Note that (3.1) can be interpreted as taking a *noisy linear* measurement of $\bigtriangledown f(\xi)$ with the measurement vector $\mathbf{v}$ and the noise being the Taylor remainder term. Importantly, due to the sparse additive form of $f$, we have $\phi_l \equiv 0, l \notin S$, implying that $\bigtriangledown f(\xi) = [\phi_1'(\xi_1) \ \phi_2'(\xi_2) \ldots \phi_d'(\xi_d)]$ is at most $k$-sparse. Hence (3.1) actually represents a noisy linear measurement of the $k$-sparse vector : $\bigtriangledown f(\xi)$. For any fixed $\xi$, we know from compressive sensing (CS) [16, 17] that $\bigtriangledown f(\xi)$ can be recovered (with high probability) using few random linear measurements[4].

This motivates the following sets of points using which we query $f$ as illustrated in Figure 1. For integers $m_x, m_v > 0$ we define

$$\mathcal{X} := \left\{ \xi_i = \frac{i}{m_x}(1,1,\ldots,1)^T \in \mathbb{R}^d : i = -m_x,\ldots,m_x \right\}, \tag{3.2}$$

$$\mathcal{V} := \left\{ v_j \in \mathbb{R}^d : v_{j,l} = \pm\frac{1}{\sqrt{m_v}} \text{ w.p. } 1/2 \text{ each}; \ j = 1,\ldots,m_v \text{ and } l = 1,\ldots,d \right\}. \tag{3.3}$$

Using (3.1) at each $\xi_i \in \mathcal{X}$ and $\mathbf{v}_j \in \mathcal{V}$ for $i = -m_x,\ldots,m_x$ and $j = 1,\ldots,m_v$ leads to:

$$\underbrace{\frac{f(\xi_i + \epsilon\mathbf{v}_j) - f(\xi_i)}{\epsilon}}_{y_{i,j}} = \langle \mathbf{v}_j, \underbrace{\bigtriangledown f(\xi_i)}_{\mathbf{x}_i} \rangle + \underbrace{\frac{1}{2}\epsilon\mathbf{v}_j^T \bigtriangledown^2 f(\zeta_{i,j})\mathbf{v}_j}_{n_{i,j}}, \tag{3.4}$$

where $\mathbf{x}_i = \nabla f(\xi_i) = [\phi_1'(i/m_x)\ \phi_2'(i/m_x) \ldots \phi_d'(i/m_x)]$ is $k$-sparse. Let us denote $\mathbf{V} = [\mathbf{v}_1 \ldots \mathbf{v}_{m_v}]^T$, $\mathbf{y}_i = [y_{i,1} \ldots y_{i,m_v}]$ and $\mathbf{n}_i = [n_{i,1} \ldots n_{i,m_v}]$. Then for each $i$, we can write (3.4) in the succinct form:

$$\mathbf{y}_i = \mathbf{V}\mathbf{x}_i + \mathbf{n}_i. \qquad (3.5)$$

Here $\mathbf{V} \in \mathbb{R}^{m_v \times d}$ represents the linear measurement matrix, $\mathbf{y}_i \in \mathbb{R}^{m_v}$ denotes the measurement vector at $\xi_i$ and $\mathbf{n}_i$ represents "noise" on account of non-linearity of $f$. Note that we query $f$ at $|\mathcal{X}|(|\mathcal{V}|+1) = (2m_x+1)(m_v+1)$ many points. Given $\mathbf{y}_i, \mathbf{V}$ we can recover a robust approximation to $\mathbf{x}_i$ via $\ell_1$ minimization [16, 17]. On account of the structure of $\nabla f$, we thus recover noisy estimates to $\phi_l'$ at equispaced points along the interval $[-1,1]$. We are now in a position to formally present our algorithm for learning SPAMs.

**Our algorithm for learning SPAMs.** The steps involved in our learning scheme are outlined in Algorithm 1. Steps 1-4 involve the CS-based recovery stage wherein we use the aforementioned sampling sets to formulate our problem as a CS one. Step 4 involves a simple thresholding procedure where an appropriate threshold $\tau$ is employed to recover the unknown active set $S$. In Section 4 we provide precise conditions on our sampling parameters which guarantee exact recovery, i.e. $\widehat{S} = S$. Step 5 leverages a convex quadratic program (P), that uses noisy estimates of $\phi_l'(i/m_x)$, i.e., $\widehat{x}_{i,l}$ for each $l \in \widehat{S}$ and $i = -m_x, \ldots, m_x$, to return a cubic spline estimate $\tilde{\phi}_l'$. This program and its theoretical properties are explained in Section 4. Finally, in Step 6 we derive our final estimate $\phi_{\text{est},l}$ via piecewise integration of $\tilde{\phi}_l'$ for each $l \in \widehat{S}$. Hence our final estimate of $\phi_l$ is a spline of degree 4. The performance of Algorithm 1 for recovering $S$ and the individual $\phi_l$'s is presented in Theorem 1, which is also our first main result. All proofs are deferred to the appendix.

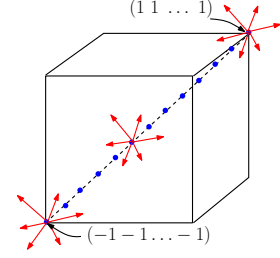

Figure 1: The points $\xi_i \in \mathcal{X}$ (blue disks) and $\xi_i + \epsilon \mathbf{v}_j$ (red arrows) for $\mathbf{v}_j \in \mathcal{V}$.

---

**Algorithm 1** Algorithm for learning $\phi_l$ in the SPAM: $f(\mathbf{x}) = \sum_{l \in S} \phi_l(x_l)$

1: Choose $m_x$, $m_v$ and construct sampling sets $\mathcal{X}$ and $\mathcal{V}$ as in (3.2), (3.3).
2: Choose step size $\epsilon > 0$. Query $f$ at $f(\xi_i), f(\xi_i + \epsilon \mathbf{v}_j)$ for $i = -m_x, \ldots, m_x$ and $j = 1, \ldots, m_v$.
3: Construct $\mathbf{y}_i$ where $y_{i,j} = \frac{f(\xi_i + \epsilon \mathbf{v}_j) - f(\xi_i)}{\epsilon}$ for $i = -m_x, \ldots, m_x$ and $j = 1, \ldots, m_v$.
4: Set $\widehat{\mathbf{x}}_i := \underset{\mathbf{y}_i = \mathbf{V}\mathbf{z}}{\operatorname{argmin}} \|\mathbf{z}\|_1$. For $\tau > 0$ compute $\widehat{S} = \cup_{i=-m_x}^{m_x} \{l \in \{1, \ldots, d\} : |\widehat{x}_{i,l}| > \tau\}$.
5: For each $l \in \widehat{S}$, run (P) as defined in Section 4 using $(\widehat{x}_{i,l})_{i=-m_x}^{m_x}$, $\tau$ and some smoothing parameter $\gamma \geq 0$, to obtain $\tilde{\phi}_l'$.
6: For each $l \in \widehat{S}$, set $\phi_{\text{est},l}$ to be the piece-wise integral of $\tilde{\phi}_l'$ as explained in Section 4.

---

**Theorem 1.** *There exist constants $C, C_1 > 0$ such that if $m_x \geq (1/\delta)$, $m_v \geq C_1 k \log d$, $0 < \epsilon < \frac{D\sqrt{m_v}}{CkB_2}$ and $\tau = \frac{C\epsilon kB_2}{2\sqrt{m_v}}$ then with high probability, $\widehat{S} = S$ and for any $\gamma \geq 0$ the estimate $\phi_{\text{est},l}$ returned by Algorithm 1 satisfies for each $l \in S$:*

$$\|\phi_{\text{est},l} - \phi_l\|_{L^\infty[-1,1]} \leq [59(1+\gamma)] \frac{C\epsilon kB_2}{\sqrt{m_v}} + \frac{87}{64m_x^4} \|\phi_l^{(5)}\|_{L^\infty[-1,1]}. \qquad (3.6)$$

Recall that $k, B_2, D, \delta$ are our problem parameters introduced in Section 2, while $\epsilon$ is the step size parameter from (3.4). We see that with $O(k \log d)$ point queries of $f$ and with $\epsilon < \frac{D\sqrt{m_v}}{CkB_2}$, the active set is recovered exactly. The error bound in (3.6) holds for all such choices of $\epsilon$. It is a sum of two terms in which the first one arises during the estimation of $\nabla f$ during the CS stage. The second error term is the interpolation error bound for interpolating $\phi_l'$ from its samples in the noise-free setting. We note that our point queries lie in $[-(1+(\epsilon/\sqrt{m_v})), (1+(\epsilon/\sqrt{m_v}))]^d$. For the stated condition on $\epsilon$ in Theorem 1 we have $\epsilon/\sqrt{m_v} < \frac{D}{CkB_2}$ which can be made arbitrarily close to zero by choosing an appropriately small $\epsilon$. Hence we sample from only a small enlargement of $[-1,1]^d$.

# 4 Analyzing the algorithm

We now describe and analyze in more detail the individual stages of Algorithm 1. We first analyze Steps 1-4 which constitute the compressive sensing (CS) based recovery stage. Next, we analyze Step 5 where we also introduce our convex quadratic program. Lastly, we analyze Step 6 where we derive our final estimate $\phi_{\text{est},l}$.

**Compressive sensing-based recovery stage.** This stage of Algorithm 1 involves solving a sequence of linear programs for recovering estimates of $\mathbf{x}_i = [\phi'_1(i/m_x) \ \ldots \ \phi'_d(i/m_x)]$ for $i = -m_x, \ldots, m_x$. We note that the measurements $\mathbf{y}_i$ are noisy linear measurements of $\mathbf{x}_i$ with the noise being arbitrary and bounded. For such a noise model, it is known that $\ell_1$ minimization results in robust recovery of the sparse signal [18]. Using this result in our setting allows us to quantify the recovery error $\| \widehat{\mathbf{x}}_i - \mathbf{x}_i \|_2$ as specified in Lemma 1.

**Lemma 1.** *There exist constants $c'_3 \geq 1$ and $C, c'_1 > 0$ such that for $m_v$ satisfying $c'_3 k \log d < m_v < d/(\log 6)^2$ we have with probability at least $1 - e^{-c'_1 m_v} - e^{-\sqrt{m_v d}}$ that $\widehat{\mathbf{x}}_i$ satisfies $\| \widehat{\mathbf{x}}_i - \mathbf{x}_i \|_2 \leq \frac{C\epsilon k B_2}{2\sqrt{m_v}}$ for all $i = -m_x, \ldots, m_x$. Furthermore, given that this holds and $m_x \geq 1/\delta$ is satisfied we then have for any $\epsilon < \frac{D\sqrt{m_v}}{CkB_2}$ that the choice $\tau = \frac{C\epsilon k B_2}{2\sqrt{m_v}}$ implies that $\widehat{S} = S$.*

Thus upon using $\ell_1$ minimization based decoding at $2m_x + 1$ points, we recover robust estimates $\widehat{\mathbf{x}}_i$ to $\mathbf{x}_i$ which immediately gives us estimates $\widehat{\phi}'_l(i/m_x) = \widehat{\mathbf{x}}_{i,l}$ of $\phi'_l(i/m_x)$ for $i = -m_x, \ldots, m_x$ and $l = 1, \ldots, d$. In order to recover the active set $S$, we first note that the spacing between consecutive samples in $\mathcal{X}$ is $1/m_x$. Therefore the condition $m_x \geq 1/\delta$ implies on account of Assumption 3 that the sample spacing is fine enough to ensure that for each $l \in S$, there exists a sample $i$ for which $|\phi'_l(i/m_x)| \geq D$ holds. The stated choice of the step size $\epsilon$ essentially guarantees $\forall l \notin S, i$ that $\left| \widehat{\phi}'_l(i/m_x) \right|$ lies within a sufficiently small neighborhood of the origin in turn enabling detection of the active set. Therefore after this stage of Algorithm 1, we have at hand: the active set $S$ along with the estimates: $(\widehat{\phi}'_l(i/m_x))_{i=m_x}^{m_x}$ for each $l \in S$. Furthermore, it is easy to see that $\left| \widehat{\phi}'_l(i/m_x) - \phi'_l(i/m_x) \right| \leq \tau = \frac{C\epsilon k B_2}{2\sqrt{m_v}}, \forall l \in S, \forall i$.

**Robust estimation via cubic splines.** Our aim now is to recover a smooth, robust estimate to $\phi'_l$ by using the noisy samples $(\widehat{\phi}'_l(i/m_x))_{i=m_x}^{m_x}$. Note that the noise here is arbitrary and bounded by $\tau = \frac{C\epsilon k B_2}{2\sqrt{m_v}}$. To this end we choose to use cubic splines as our estimates, which are essentially piecewise cubic polynomials that are $C^2$ smooth [19]. There is a considerable amount of literature in the statistics community devoted to the problem of estimating univariate functions from noisy samples via cubic splines (cf., [20, 21, 22, 23]), albeit under the setting of random noise. Cubic splines have also been studied extensively in the approximation theoretic setting for *interpolating* samples (cf., [19, 24, 25]).

We introduce our solution to this problem in a more general setting. Consider a smooth function $g : [t_1, t_2] \to \mathbb{R}$ and a uniform mesh[5]: $\prod : t_1 = x_0 < x_1 < \cdots < x_{n-1} < x_n = t_2$ with $x_i - x_{i-1} = h$. We have at hand noisy samples: $\widehat{g}_i = g(x_i) + e_i$, with noise $e_i$ being arbitrary and bounded: $|e_i| \leq \tau$. In the noiseless scenario, the problem would be an interpolation one for which a popular class of cubic splines are the "not-a-knot" cubic splines [24]. These achieve optimal $O(h^4)$ error rates for $C^4$ smooth $g$ without using any higher order information about $g$ as boundary conditions. Let $H^2[t_1, t_2]$ denote the space of cubic splines defined on $[t_1, t_2]$ w.r.t $\prod$. We then propose finding the cubic spline estimate as a solution of the following convex optimization problem (in the $4n$ coefficients of the $n$ cubic polynomials) for some parameter $\gamma \geq 0$:

$$
(\text{P}) \begin{cases} \min_{\mathcal{L} \in H^2[t_1, t_2]} \int_{t_1}^{t_2} \mathcal{L}''(x)^2 dx & (4.1) \\ \text{s.t.} \quad \widehat{g}_i - \gamma\tau \leq \mathcal{L}(x_i) \leq \widehat{g}_i + \gamma\tau; \quad i = 0, \ldots, n, & (4.2) \\ \mathcal{L}'''(x_1^-) = \mathcal{L}'''(x_1^+), \quad \mathcal{L}'''(x_{n-1}^-) = \mathcal{L}'''(x_{n-1}^+). & (4.3) \end{cases}
$$

Note that (P) is a convex QP with linear constraints. The objective function can be verified to be a positive definite quadratic form in the spline coefficients[6]. Specifically, the objective measures the total curvature of a feasible cubic spline in $[t_1, t_2]$. Each of the constraints (4.2)-(4.3) along with the implicit continuity constraints of $\mathcal{L}^{(p)}$; $p = 0, 1, 2$ at the interior points of $\prod$, are linear equalities/inequalities in the coefficients of the piecewise cubic polynomials. (4.3) refers to the not-a-knot boundary conditions [24] which are also linear equalities in the spline coefficients. These conditions imply that $\mathcal{L}'''$ is continuous[7] at the knots $x_1, x_{n-1}$. Thus, (P) searches amongst the space of all not-a-knot cubic splines such that $\mathcal{L}(x_i)$ lies within a $\pm\gamma\tau$ interval of $\widehat{g}_i$, and returns the smoothest solution, i.e., the one with the least total curvature. The parameter $\gamma \geq 0$, controls the degree of smoothness of the solution. Clearly, $\gamma = 0$ implies interpolating the noisy samples $(\widehat{g}_i)_{i=0}^n$. As $\gamma$ increases, the search interval: $[\widehat{g}_i - \gamma\tau, \widehat{g}_i + \gamma\tau]$ becomes larger for all $i$, leading to smoother feasible cubic splines. The following theorem formally describes the estimation properties of (P) and is also our second main result.

**Theorem 2.** *For $g \in C^4[t_1, t_2]$ let $\mathcal{L}^* : [t_1, t_2] \to \mathbb{R}$ be a solution of (P) for some parameter $\gamma \geq 0$. We then have that*

$$\| \mathcal{L}^* - g \|_\infty \leq \left[\frac{118(1 + \gamma)}{3}\right]\tau + \frac{29}{64}h^4 \| g^{(4)} \|_\infty . \qquad (4.4)$$

We show in the appendix that if $\int_{t_1}^{t_2}(\mathcal{L}^{*''}(x))^2 dx > 0$, then $\mathcal{L}^*$ is unique. Note that the error bound (4.4) is a sum of two terms. The first term is proportional to the external noise bound: $\tau$, indicating that the solution is *robust* to noise. The second term is the error that would arise even if perturbation was absent i.e. $\tau = 0$. Intuitively, if $\gamma\tau$ is large enough, then we would expect the solution returned by (P) to be a line. Indeed, a larger value of $\gamma\tau$ would imply a larger search interval in (4.2), which if sufficiently large, would allow a line (that has zero curvature) to lie in the feasible region. More formally, we show in the appendix, sufficient conditions: $\tau = \Omega(\frac{n^{1/2}\|g''\|_\infty}{\gamma-1})$, $\gamma > 1$, which if satisfied, imply that the solution returned by (P) is a line. This indicates that if either $n$ is small or $g$ has small curvature, then moderately large values of $\tau$ and/or $\gamma$ will cause the solution returned by (P) to be a line. If an estimate of $\| g'' \|_\infty$ is available, then one could for instance, use the upper bound $1 + O(n^{1/2} \| g'' \|_\infty /\tau)$ to restrict the range of values of $\gamma$ within which (P) is used.

Theorem 2 has the following Corollary for estimation of $C^4$ smooth $\phi_l'$ in the interval $[-1, 1]$. The proof simply involves replacing: $g$ with $\phi_l'$, $n + 1$ with $2m_x + 1$, $h$ with $1/m_x$ and $\tau$ with $\frac{C\epsilon k B_2}{2\sqrt{m_v}}$. As the perturbation $\tau$ is directly proportional to the step size $\epsilon$, we show in the appendix that if additionally $\epsilon = \Omega(\frac{\sqrt{m_x m_v}\|\phi_l''\|_\infty}{\gamma-1})$, $\gamma > 1$, holds, then the corresponding estimate $\tilde{\phi}'_l$ will be a line.

**Corollary 1.** *Let (P) be employed for each $l \in S$ using noisy samples $\left\{\widehat{\phi}'_l(i/m_x)\right\}_{i=-m_x}^{m_x}$, and with step size $\epsilon$ satisfying $0 < \epsilon < \frac{D\sqrt{m_v}}{CkB_2}$. Denoting $\tilde{\phi}'_l$ as the corresponding solution returned by (P), we then have for any $\gamma \geq 0$ that:*

$$\| \tilde{\phi}'_l - \phi'_l \|_{L^\infty[-1,1]} \leq \left[\frac{59(1 + \gamma)}{3}\right]\frac{C\epsilon k B_2}{\sqrt{m_v}} + \frac{29}{64m_x^4} \| \phi_l^{(5)} \|_{L^\infty[-1,1]} . \qquad (4.5)$$

**The final estimate.** We now derive the final estimate $\phi_{\text{est},l}$ of $\phi_l$ for each $l \in S$. Denote $x_0(= -1) < x_1 < \cdots < x_{2m_x-1} < x_{2m_x}(= 1)$ as our equispaced set of points on $[-1, 1]$. Since $\tilde{\phi}'_l : [-1, 1] \to \mathbb{R}$ returned by (P) is a cubic spline, we have $\tilde{\phi}'_l(x) = \tilde{\phi}'_{l,i}(x)$ for $x \in [x_i, x_{i+1}]$ where $\tilde{\phi}'_{l,i}$ is a polynomial of degree at most 3. We then define $\phi_{\text{est},l}(x) := \tilde{\phi}_{l,i}(x) + F_i$ for $x \in [x_i, x_{i+1}]$ and $i = 0, \ldots, 2m_x - 1$. Here $\tilde{\phi}_{l,i}$ is a antiderivative of $\tilde{\phi}'_{l,i}$ and $F_i$'s are constants of integration. Denoting $F_0 = F$, we have that $\phi_{\text{est},l}$ is continuous at $x_1, \ldots, x_{2m_x-1}$ for: $F_i = \tilde{\phi}_{l,0}(x_1) + \sum_{j=1}^{i-1}(\tilde{\phi}_{l,j}(x_{j+1}) - \tilde{\phi}_{l,j}(x_j)) - \tilde{\phi}_{l,i}(x_i) + F = F_i' + F$; $1 \leq i \leq 2m_x - 1$. Hence by denoting $\psi_{l,i}(\cdot) := \tilde{\phi}_{l,i}(\cdot) + F_i'$ we obtain $\phi_{\text{est},l}(\cdot) = \psi_l(\cdot) + F$ where $\psi_l(x) = \psi_{l,i}(x)$ for

$x \in [x_i, x_{i+1}]$. Now on account of Assumption 2, we require $\phi_{\mathrm{est},l}$ to also be centered implying $F = -\frac{1}{2}\int_{-1}^{1}\psi_l(x)dx$. Hence we output our final estimate of $\phi_l$ to be:

$$\phi_{\mathrm{est},l}(x) := \psi_l(x) - \frac{1}{2}\int_{-1}^{1}\psi_l(x)dx; \quad x \in [-1,1]. \tag{4.6}$$

Since $\phi_{\mathrm{est},l}$ is by construction continuous in $[-1,1]$, is a piecewise combination of polynomials of degree at most 4, and since $\phi'_{\mathrm{est},l}$ is a cubic spline, $\phi_{\mathrm{est},l}$ is a spline function of order 4. Lastly, we show in the proof of Theorem 1 that $\| \phi_{\mathrm{est},l} - \phi_l \|_{L^{\infty}[-1,1]} \leq 3 \| \tilde{\phi}'_l - \phi'_l \|_{L^{\infty}[-1,1]}$ holds. Using Corollary 1, this provides us with the error bounds stated in Theorem 1.

## 5  Impact of noise on performance of our algorithm

Our third main contribution involves analyzing the more realistic scenario, when the point queries are corrupted with additive external noise $z'$. Thus querying $f$ in Step 2 of Algorithm 1 results in noisy values: $f(\xi_i) + z'_i$ and $f(\xi_i + \epsilon\mathbf{v}_j) + z'_{i,j}$ respectively. This changes (3.5) to the noisy linear system: $\mathbf{y}_i = \mathbf{V}\mathbf{x}_i + \mathbf{n}_i + \mathbf{z}_i$ where $z_{i,j} = (z'_{i,j} - z'_i)/\epsilon$ for $i = -m_x, \ldots, m_x$ and $j = 1, \ldots, m_v$. Notice that external noise gets scaled by $(1/\epsilon)$, while $|n_{i,j}|$ scales linearly with $\epsilon$.

**Arbitrary bounded noise.** In this model, the external noise is arbitrary but bounded, so that $|z'_i|, |z'_{i,j}| < \kappa; \forall i, j$. It can be verified along the lines of the proof of Lemma 1 that: $\| \mathbf{n}_i + \mathbf{z}_i \|_2 \leq \sqrt{m_v}\left(\frac{2\kappa}{\epsilon} + \frac{\epsilon k B_2}{2m_v}\right)$. Observe that unlike the noiseless setting, $\epsilon$ cannot be made arbitrarily close to 0, as it would blow up the impact of the external noise. The following theorem shows that if $\kappa$ is small relative to $D^2 < |\phi'_l(x)|^2, \forall x \in I_l, l \in S$, then[8] there exists an interval for choosing $\epsilon$, within which Algorithm 1 recovers exactly the active set $S$. This condition has the natural interpretation that if the signal-to-'external noise' ratio in $I_l$ is sufficiently large, then $S$ can be detected exactly.

**Theorem 3.** *There exist constants $C, C_1 > 0$ such that if $\kappa < D^2/(16C^2kB_2)$, $m_x \geq (1/\delta)$, and $m_v \geq C_1 k \log d$ hold, then for any $\epsilon \in \frac{D\sqrt{m_v}}{2CkB_2}[1 - A, 1 + A]$ where $A := \sqrt{1 - (16C^2kB_2\kappa)/D^2}$ and $\tau = \sqrt{m_v}\left(\frac{2\kappa}{\epsilon} + \frac{\epsilon k B_2}{2m_v}\right)$, we have in Algorithm 1, with high probability, that $\widehat{S} = S$ and for any $\gamma \geq 0$, for each $l \in S$:*

$$\| \phi_{est,l} - \phi_l \|_{L^{\infty}[-1,1]} \leq [59(1+\gamma)]\left(\frac{4C\sqrt{m_v}\kappa}{\epsilon} + \frac{C\epsilon k B_2}{\sqrt{m_v}}\right) + \frac{87}{64m_x^4} \| \phi_l^{(5)} \|_{L^{\infty}[-1,1]}. \tag{5.1}$$

**Stochastic noise.** In this model, the external noise is assumed to be i.i.d. Gaussian, so that $z'_i, z'_{i,j} \sim \mathcal{N}(0, \sigma^2)$; i.i.d. $\forall i, j$. In this setting we consider resampling $f$ at the query point $N$ times and then averaging the noisy samples, in order to reduce $\sigma$. Given this, we now have that $z'_i, z'_{i,j} \sim \mathcal{N}(0, \frac{\sigma^2}{N})$; i.i.d. $\forall i, j$. Using standard tail-bounds for Gaussians, we can show that for any $\kappa > 0$ if $N$ is chosen large enough then: $|z_{i,j}| = |z'_i - z'_{i,j}| \leq 2\kappa; \forall i, j$ with high probability. Hence the external noise $z_{i,j}$ would be bounded with high probability and the analysis for Theorem 3 can be used in a straightforward manner. Of course, an advantage that we have in this setting is that $\kappa$ can be chosen to be arbitrarily close to zero by choosing a correspondingly large value of $N$. We state all this formally in the form of the following theorem.

**Theorem 4.** *There exist constants $C, C_1 > 0$ such that for $\kappa < D^2/(16C^2kB_2)$, $m_x \geq (1/\delta)$, and $m_v \geq C_1 k \log d$, if we re-sample each query in Step 2 of Algorithm 1: $N > \frac{\sigma^2}{\kappa^2}\log\left(\frac{\sqrt{2}\sigma}{\kappa p}|\mathcal{X}||\mathcal{V}|\right)$ times for $0 < p < 1$, and average the values, then for any $\epsilon \in \frac{D\sqrt{m_v}}{2CkB_2}[1 - A, 1 + A]$ where $A := \sqrt{1 - (16C^2kB_2\kappa)/D^2}$ and $\tau = \sqrt{m_v}\left(\frac{2\kappa}{\epsilon} + \frac{\epsilon k B_2}{2m_v}\right)$, we have in Algorithm 1, with probability at least $1 - p - o(1)$, that $\widehat{S} = S$ and for any $\gamma \geq 0$, for each $l \in S$:*

$$\| \phi_{est,l} - \phi_l \|_{L^{\infty}[-1,1]} \leq [59(1+\gamma)]\left(\frac{4C\sqrt{m_v}\kappa}{\epsilon} + \frac{C\epsilon k B_2}{\sqrt{m_v}}\right) + \frac{87}{64m_x^4} \| \phi_l^{(5)} \|_{L^{\infty}[-1,1]}. \tag{5.2}$$

Note that we query $f$ now $N |\mathcal{X}| (|\mathcal{V}| + 1)$ times. Also, $|\mathcal{X}| = (2m_x + 1) = \Theta(1)$, and $\kappa = O(k^{-1})$, as $D, C, B_2, \delta$ are constants. Hence the choice $|\mathcal{V}| = O(k \log d)$ gives us $N = O(k^2 \log(p^{-1}k^2 \log d))$ and leads to an overall query complexity of: $O(k^3 \log d \log(p^{-1}k^2 \log d))$ when the samples are corrupted with additive Gaussian noise. Choosing $p = O(d^{-c})$ for any constant $c > 0$ gives us a sample complexity of $O(k^3(\log d)^2)$, and ensures that the result holds with high probability. The $o(1)$ term goes to zero exponentially fast as $d \to \infty$.

**Simulation results.** We now provide simulation results on synthetic data to support our theoretical findings. We consider the noisy setting with the point queries being corrupted with Gaussian noise. For $d = 1000, k = 4$ and $S = \{2, 105, 424, 782\}$, consider $f : \mathbb{R}^d \to \mathbb{R}$ where $f = \phi_2(x_2) + \phi_{105}(x_{105}) + \phi_{424}(x_{424}) + \phi_{782}(x_{782})$ with: $\phi_2(x) = \sin(\pi x)$, $\phi_{105}(x) = \exp(-2x)$, $\phi_{424}(x) = (1/3)\cos^3(\pi x) + 0.8x^2$, $\phi_{782}(x) = 0.5x^4 - x^2 + 0.8x$. We choose $\delta = 0.3, D = 0.2$ which can be verified as valid parameters for the above $\phi_l$'s. Furthermore, we choose $m_x = \lceil 2/\delta \rceil = 7$ and $m_v = \lceil 2k \log d \rceil = 56$ to satisfy the conditions of Theorem 4. Next, we choose constants $C = 0.2, B_2 = 35$ and $\kappa = 0.95 \frac{D^2}{16C^2 k B_2} = 4.24 \times 10^{-4}$ as required by Theorem 4. For the choice $\epsilon = \frac{D\sqrt{m_v}}{2Ck B_2} = 0.0267$, we then query $f$ at $(2m_x + 1)(m_v + 1) = 855$ points. The function values are corrupted with Gaussian noise: $\mathcal{N}(0, \sigma^2/N)$ for $\sigma = 0.01$ and $N = 100$. This is equivalent to resampling and averaging the points queries $N$ times. Importantly the sufficient condition on $N$, as stated in Theorem 4 is $\lceil \frac{\sigma^2}{\kappa^2} \log(\frac{\sqrt{2}\sigma|\mathcal{X}||\mathcal{V}|}{\kappa p}) \rceil = 6974$ for $p = 0.1$. Thus we consider a significantly *undersampled* regime. Lastly we select the threshold $\tau = \sqrt{m_v}\left(\frac{2\kappa}{\epsilon} + \frac{\epsilon k B_2}{2m_v}\right) = 0.2875$ as stated by Theorem 4, and employ Algorithm 1 for different values of the smoothing parameter $\gamma$.

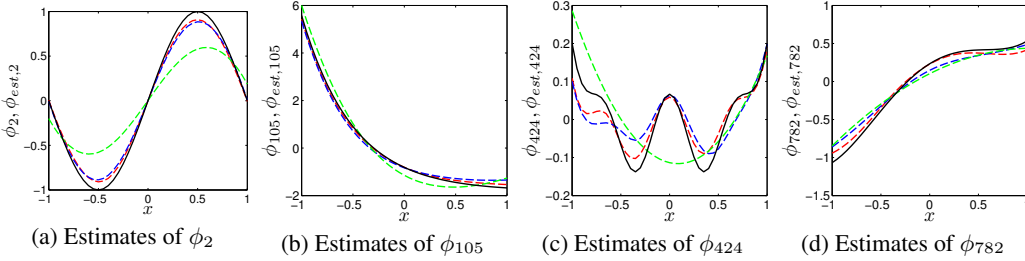

(a) Estimates of $\phi_2$     (b) Estimates of $\phi_{105}$     (c) Estimates of $\phi_{424}$     (d) Estimates of $\phi_{782}$

Figure 2: Estimates $\phi_{\text{est},l}$ of $\phi_l$ (black) for: $\gamma = 0.3$ (red), $\gamma = 1$ (blue) and $\gamma = 5$ (green).

The results are shown in Figure 2. Over 10 independent runs of the algorithm we observed that $S$ was recovered exactly each time. Furthermore we see from Figure 2 that the recovery is quite accurate for $\gamma = 0.3$. For $\gamma = 1$ we notice that the search interval $\gamma\tau = 0.2875$ becomes large enough so as to cause the estimates $\phi_{\text{est},424}, \phi_{\text{est},782}$ to become relatively smoother. For $\gamma = 5$, the search interval $\gamma\tau = 1.4375$ becomes wide enough for a line to fit in the feasible region for $\phi'_{424}, \phi'_{782}$. This results in $\phi_{\text{est},424}, \phi_{\text{est},782}$ to be quadratic functions. In the case of $\phi'_2, \phi'_{105}$, the search interval is not sufficiently wide enough for a line to lie in the feasible region, even for $\gamma = 5$. However we notice that the estimates $\phi_{\text{est},2}, \phi_{\text{est},105}$ become relatively smoother as expected.

## 6  Conclusion

We proposed an efficient sampling scheme for learning SPAMs. In particular, we showed that with only a few queries, we can derive uniform approximations to each underlying univariate function of the SPAM. A crucial component of our approach is a novel convex QP for robust estimation of univariate functions via cubic splines, from samples corrupted with arbitrary bounded noise. Lastly, we showed how our algorithm can handle noisy point queries for both (i) arbitrary bounded and (ii) i.i.d. Gaussian noise models. An important direction for future work would be to determine the optimality of our sampling bounds by deriving corresponding lower bounds on the sample complexity.

**Acknowledgments.** This research was supported in part by SNSF grant 200021_137528 and a Microsoft Research Faculty Fellowship.

## Footnotes

[1] $\| f \|^2_{L^2(\mathbb{P})} = \int |f(\mathbf{x})|^2 \, d\mathbb{P}(\mathbf{x})$ and $\| f \|^2_{L^2(\mathbb{P}_n)} = \frac{1}{n} \sum_i f^2(\mathbf{x}_i)$

[2] This means in the $L^\infty$ norm

[3] In case $f : [a,b]^d \to \mathbb{R}$ we can define $g : [-1,1]^d \to \mathbb{R}$ where $g(\mathbf{x}) = f(\frac{(b-a)}{2}\mathbf{x} + \frac{b+a}{2}) = \sum_{l \in S} \tilde{\phi}_l(x_l)$ with $\tilde{\phi}_l(x_l) = \phi_l(\frac{(b-a)}{2}x_l + \frac{b+a}{2})$. We then sample $g$ from within $[-(1+r),(1+r)]^d$ for some small $r > 0$ by querying $f$, and estimate $\tilde{\phi}_l$ in $[-1,1]$ which in turn gives an estimate to $\phi_l$ in $[a,b]$.

[4] Estimating sparse gradients via compressive sensing has been considered previously by Fornasier et al. [8] albeit for a substantially different function class than us. Hence their sampling scheme differs considerably from ours, and is not tailored for learning SPAMs.

[5]We consider uniform meshes for clarity of exposition. The results in this section can be easily generalized to non-uniform meshes.

[6]Shown in the appendix.

[7]$f(x^-) = \lim_{h\to 0^-} f(x + h)$ and $f(x^+) = \lim_{h\to 0^+} f(x + h)$ denote left,right hand limits respectively.

[8] $I_l$ is the "critical" interval defined in Assumption 3 for detecting $l \in S$.

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
