[Supplementary Material]

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

[9] For ease of exposition we will restrict ourselves to uniform meshes in this paper. However one can consider a non-uniform mesh too.

[10] Actually "C. de Boor, Convergence of cubic spline interpolation with the not-a-knot condition,Technical report,1984" in which this was proven contains a bound for general meshes (not necessarily uniform.)

[11] The constant $29/64$ can most likely be improved using the integral form of Taylors remainder term.

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

## Supplementary Material : Efficient Sampling for Learning Sparse Additive Models in High Dimensions.

In this supplementary material, we prove the results stated in the paper. We first provide a collection of general results along with proofs in Section A. which are used in order to prove our main results. Section B contains the proof of Lemma 1 while in Section C we prove Theorem 2. Section D contains the proof of our main theorem for estimating in the the absence of external noise in point queries, i.e. Theorem 1. In Section E we prove Theorem 3, which is concerned with the setting where the point queries are corrupted by arbitrary bounded noise. In Section F we prove Theorem 4 that handles the setting of stochastic noise corrupting the point queries.

## A    Some general results

The following Proposition is well known.

**Proposition 1.** *Let* $g \in C^2[x_i, x_{i+1}]$ *be such that* $g(x_i) = g(x_{i+1}) = 0$. *We then have that* $\| g \|_\infty \leq (1/8) \| g^{(2)} \|_\infty h^2$ *where* $h := x_{i+1} - x_i$.

*Proof.* Consider $e(x) = g(x) - w(t)(x - x_i)(x - x_{i+1})$ where for any fixed $t \in (x_i, x_{i+1})$, $w(t)$ is chosen so that

$$w(t) = \frac{g(t)}{(t - x_i)(t - x_{i+1})}. \tag{A.1}$$

Since $t \neq x_i, x_{i+1}$, $w(t)$ is well defined. Hence $e(x_i) = e(x_{i+1}) = e(t) = 0$. By repeated application of Rolle's theorem, this then implies that $\exists \zeta \in (x_i, x_{i+1})$ such that $e''(\zeta) = 0$. But $e''(x) = g''(x) - 2w(t)$, hence $w(t) = g''(\zeta)/2$. Lastly we note that $|(t - x_i)(t - x_{i+1})| \leq \frac{h^2}{4}$. Lastly, since $e(t) = 0$ we obtain

$$g(t) = \frac{g''(\zeta)}{2}(t - x_i)(t - x_{i+1}) \Rightarrow |g(t)| \leq \frac{\| g'' \|_\infty h^2}{8}. \tag{A.2}$$

□

The following lemma is a simplified version of Lemma 1 of "T.Lucas, Error Bounds for Interpolating Cubic Splines Under Various End Conditions,SIAM Journal on Num. Anal.,1974".

**Lemma 2.** *Let* $f \in C^4[t_1, t_2]$ *and let* $\mathcal{L}$ *be a cubic spline interpolate of* $f$ *on* $[t_1, t_2]$ *on the mesh* $(t_1 =)x_0 < x_1 < \cdots < x_n(= t_2)$ *with* $x_i - x_{i-1} = h$ *for* $i = 1, \ldots, n$. *If*

$$\max_{0 \leq i \leq n} |f_i'' - \mathcal{L}_i''| \leq Kh^2 \tag{A.3}$$

*holds for some constant* $K > 0$ *then it implies that :*

$$\| f - \mathcal{L} \|_\infty \leq \frac{h^4}{8} \left( K + \frac{\| f^{(4)} \|_\infty}{8} \right). \tag{A.4}$$

*Proof.* Since $\mathcal{L}''$ is linear for $x \in [x_i, x_{i+1}]$ we have:

$$\mathcal{L}''(x) = \frac{x - x_i}{h} \mathcal{L}_{i+1}'' + \frac{(x_{i+1} - x)}{h} \mathcal{L}_i''; \quad x \in [x_i, x_{i+1}]. \tag{A.5}$$

Similarly we obtain

$$f''(x) = \underbrace{\frac{x - x_i}{h} f_{i+1}'' + \frac{(x_{i+1} - x)}{h} f_i''}_{l_{f''}(x)} + R_i(x); \quad x \in [x_i, x_{i+1}] \tag{A.6}$$

where $l_{f''}(x)$ is the linear intepolant of $f''$ between $x_i$ and $x_{i+1}$. Since $f'' - l_{f''} \in C^2[x_i, x_{i+1}]$ and $f''(x_i) - l_{f''}(x_i) = f''(x_{i+1}) - l_{f''}(x_{i+1}) = 0$, therefore by Proposition 1 we have that $|R_i(x)| \leq (1/8)h^2 \| f^{(4)} \|_\infty$. Subtracting (A.5) from (A.6) we then obtain

$$\| f'' - \mathcal{L}'' \|_\infty \le \max_{0 \le i \le n} |f_i'' - \mathcal{L}_i''| + \frac{1}{8} h^2 \| f^{(4)} \|_\infty, \tag{A.7}$$

$$\le K h^2 + \frac{1}{8} h^2 \| f^{(4)} \|_\infty = h^2 \left( K + \frac{\| f^{(4)} \|_\infty}{8} \right). \tag{A.8}$$

Since $f - \mathcal{L} \in C^2[t_1, t_2]$ and $f(x_i) - \mathcal{L}(x_i) = f(x_{i+1}) - \mathcal{L}(x_{i+1}) = 0$ for $i = 0, \ldots, n-1$ we then have from Proposition 1 that:

$$\| f - \mathcal{L} \|_\infty \le \frac{h^2}{8} \| f'' - \mathcal{L}'' \|_\infty \le \frac{h^4}{8} \left( K + \frac{\| f^{(4)} \|_\infty}{8} \right).$$

$\square$

Proposition 2 is a standard result, see for example Lemma 4 of "T.Lucas, Error Bounds for Interpolating Cubic Splines Under Various End Conditions,SIAM Journal on Num. Anal.,1974".

**Proposition 2.** *For the tridiagonal, symmetric $m \times m$ matrix*

$$\mathbf{A} = \begin{pmatrix} 4 & 1 & & & & \\ 1 & 4 & 1 & & & \\ & \ddots & \ddots & \ddots & & \\ & & \ddots & \ddots & \ddots & \\ & & & 1 & 4 & 1 \\ & & & & 1 & 4 \end{pmatrix},$$

*we have that $\mathbf{A}$ is invertible and $\| \mathbf{A}^{-1} \|_\infty \le 1/2$.*

*Proof.* The fact that $\mathbf{A}$ is invertible follows from the fact that it is strictly diagonally dominant. Now note that $\mathbf{A} = 4\mathbf{I} + \mathbf{B}$ where $\| \mathbf{B} \|_\infty \le 2$. For any $\mathbf{x} \in \mathbb{R}^m$ such that $\| \mathbf{x} \|_\infty = 1$ we have:

$$\| \mathbf{A}\mathbf{x} \|_\infty \ge 4 - \| \mathbf{B}\mathbf{x} \|_\infty \ge 4 - \| \mathbf{B} \|_\infty \ge 2. \tag{A.9}$$

Now for any $\mathbf{x}$, $\| \mathbf{x} \|_\infty = 1$; we have $\| \mathbf{A}^{-1}\mathbf{A}\mathbf{x} \|_\infty = 1$ which implies

$$\| \mathbf{A}^{-1} \left( \frac{\mathbf{A}\mathbf{x}}{\| \mathbf{A}\mathbf{x} \|_\infty} \right) \|_\infty = \frac{1}{\| \mathbf{A}\mathbf{x} \|_\infty} \le (1/2).$$

$\square$

**Proposition 3.**     1. *Let $g \in C^4[x - h_0, x + h_0]$, $h_0 > 0$. Then $\forall h \in (0, h_0)$:*

$$\left| \frac{g(x+h) - 2g(x) + g(x-h)}{h^2} - g'(x) \right| \le \frac{h^2}{12} \| g^{(4)} \|_\infty . \tag{A.10}$$

2. *Let $g \in C^2[x - h_0, x + h_0]$, $h_0 > 0$. Then $\forall h \in (0, h_0)$, $\exists \zeta \in (x - h_0, x + h_0)$ such that*

$$g(x+h) - 2g(x) + g(x-h) = h^2 g''(\zeta). \tag{A.11}$$

*Proof.*     1. We have by Taylor's expansion:

$$g(x+h) = g(x) + hg'(x) + \frac{h^2}{2}g''(x) + \frac{h^3}{6}g'''(x) + \frac{h^4}{24}g^{(4)}(\zeta_1),$$

$$g(x-h) = g(x) - hg'(x) + \frac{h^2}{2}g''(x) - \frac{h^3}{6}g'''(x) + \frac{h^4}{24}g^{(4)}(\zeta_2),$$

where $\zeta_1 \in (x, x+h)$ and $\zeta_2 \in (x-h, x)$. This then gives us:

$$\frac{g(x+h) - 2g(x) + g(x-h)}{h^2} = \frac{h^2 g''(x) + \frac{h^4}{24}g^{(4)}(\zeta_1) + \frac{h^4}{24}g^{(4)}(\zeta_2)}{h^2}$$

$$= g''(x) + \frac{h^2}{12}g^{(4)}(\zeta)$$

for some $\zeta \in (x - h_0, x + h_0)$. The second equality above follows from continuity of $g^{(4)}$ in $[x - h_0, x + h_0]$. This completes the proof.

2. We have by Taylor's expansion:

$$g(x + h) = g(x) + hg'(x) + \frac{h^2}{2}g''(\zeta_1),$$

$$g(x - h) = g(x) - hg'(x) + \frac{h^2}{2}g''(\zeta_2),$$

where $\zeta_1 \in (x, x+h)$ and $\zeta_2 \in (x-h, x)$. This then gives us for some $\zeta \in (x-h_0, x+h_0)$:

$$g(x + h) - 2g(x) + g(x - h) = h^2\frac{g''(\zeta_1) + g''(\zeta_2)}{2} = h^2 g''(\zeta)$$

where the last equality follows from continuity of $g''$.

$\square$

**Lemma 3.** *Let $f \in C^4[t_1, t_2]$ and let $(t_1 =) x_0 < x_1 < \cdots < x_n(= t_2)$ be a uniform mesh with $x_i - x_{i-1} = h$ for $i = 1, \ldots, n$. Furthermore let*

$$R_i := (f''_{i-1} + 4f''_i + f''_{i+1}) - \frac{6}{h^2}(f_{i-1} - 2f_i + f_{i+1}); \ i = 1, \ldots, n - 1. \tag{A.12}$$

*We then have $|R_i| \leq \frac{3h^2}{2} \| f^{(4)} \|_\infty, \forall i$.*

*Proof.* From Proposition 3(1) we have $\frac{f_{i-1} - 2f_i + f_{i+1}}{h^2} = f''_i + e_i$ where $|e_i| \leq (h^2/12) \| f^{(4)} \|_\infty$ for $i = 1, \ldots, n - 1$. This then gives us:

$$R_i = (f''_{i-1} - 2f''_i + f''_{i+1}) - 6e_i = h^2 f^{(4)}(\zeta_i) - 6e_i$$

where the last equality follows from Proposition 3(2) for some $\zeta_i \in (x_{i-1}, x_{i+1})$. Hence we obtain $|R_i| \leq h^2 \| f^{(4)} \|_\infty + 6 |e_i| \leq \frac{3h^2}{2} \| f^{(4)} \|_\infty$ for all $i = 1, \ldots, n - 1$. $\square$

A slightly generalized version of Proposition 4 was proven in Lemma 2 of "R.Beatson, On the Convergence of Some Cubic Spline Interpolation Schemes, SIAM Journal on Num. Anal.,1986", however the constants were not derived explicitly. In the following, we explicitly state the constants appearing in the bounds.

**Proposition 4.** *Consider the $(n + 1) \times (n + 1)$ matrix:*

$$\mathbf{A} = \begin{pmatrix} 1 & -2 & 1 & & & & \\ 1 & 4 & 1 & & & & \\ & 1 & 4 & 1 & & & \\ & & \ddots & \ddots & \ddots & & \\ & & & 1 & 4 & 1 \\ & & & & 1 & -2 & 1 \end{pmatrix}.$$

*We have that $\mathbf{A}$ is invertible and also that $\| \mathbf{A}^{-1} \|_\infty \leq (7/3)$.*

*Proof.* For some $\mathbf{c} = (c_0 \ c_1 \ \cdots \ c_{n-1} \ c_n) \in \mathbb{R}^{n+1}$ consider the linear system $\mathbf{Ax} = \mathbf{c}$ where $\mathbf{x} = (x_0 \ x_1 \ \cdots \ x_{n-1} \ x_n)$. Next, consider the first 3 rows (denoted $R_1, R_2, R_3$ from top to bottom) of the augmented matrix $[\mathbf{A} : \mathbf{c}]$:

$$\begin{pmatrix} 1 & -2 & 1 & 0 & \cdots & \cdots & 0 & \vdots & c_0 \\ 1 & 4 & 1 & 0 & \cdots & \cdots & 0 & \vdots & c_1 \\ 0 & 1 & 4 & 1 & \cdots & \cdots & 0 & \vdots & c_2 \end{pmatrix}.$$

Perform row operation $R_2 \to R_2 - R_1$ followed by $R_3 \to R_3 - (1/6)R_2$ to obtain:

$$\begin{pmatrix} 1 & -2 & 1 & 0 & \cdots & \cdots & 0 & \vdots & c_0 \\ 0 & 6 & 0 & 0 & \cdots & \cdots & 0 & \vdots & c_1 - c_0 \\ 0 & 0 & 4 & 1 & \cdots & \cdots & 0 & \vdots & c_2 - \frac{(c_1 - c_0)}{6} \end{pmatrix}.$$

Similary, consider the bottom 3 rows (denoted $R_1, R_2, R_3$ from top to bottom) of $[\mathbf{A} : \mathbf{c}]$:

$$
\begin{pmatrix}
0 & \cdots & \cdots & 1 & 4 & 1 & 0 & \vdots & c_{n-2} \\
0 & \cdots & \cdots & 0 & 1 & 4 & 1 & \vdots & c_{n-1} \\
0 & \cdots & \cdots & 0 & 1 & -2 & 1 & \vdots & c_n
\end{pmatrix}.
$$

Perform row operation $R_2 \to R_2 - R_3$ followed by $R_1 \to R_1 - (1/6)R_2$ to obtain:

$$
\begin{pmatrix}
0 & \cdots & \cdots & 1 & 4 & 0 & 0 & \vdots & c_{n-2} - \frac{(c_{n-1}-c_n)}{6} \\
0 & \cdots & \cdots & 0 & 0 & 6 & 0 & \vdots & c_{n-1} - c_n \\
0 & \cdots & \cdots & 0 & 1 & -2 & 1 & \vdots & c_n
\end{pmatrix}.
$$

We hence obtain the following equivalent system of linear equations:

$$
\underbrace{\begin{pmatrix}
4 & 1 & & & & \\
1 & 4 & 1 & & & \\
 & \ddots & \ddots & \ddots & & \\
 & & \ddots & \ddots & \ddots & \\
 & & & 1 & 4 & 1 \\
 & & & & 1 & 4
\end{pmatrix}}_{\mathbf{A}'}
\underbrace{\begin{pmatrix}
x_2 \\ x_3 \\ \vdots \\ \vdots \\ x_{n-3} \\ x_{n-2}
\end{pmatrix}}_{\mathbf{x}'}
=
\underbrace{\begin{pmatrix}
c_2 - \frac{(c_1-c_0)}{6} \\ c_3 \\ \vdots \\ \vdots \\ c_{n-3} \\ c_{n-2} - \frac{(c_{n-1}-c_n)}{6}
\end{pmatrix}}_{\mathbf{c}'} ,
\tag{A.13}
$$

where $\mathbf{A}' \in \mathbb{R}^{(n-1)\times(n-1)}$ is symmetric,tridiagonal,

$$6x_1 = c_1 - c_0 \quad \text{and} \quad x_0 - 2x_1 + x_2 = c_0, \tag{A.14}$$

$$6x_{n-1} = c_{n-1} - c_n \quad \text{and} \quad x_{n-2} - 2x_{n-1} + x_n = c_n. \tag{A.15}$$

Now $\mathbf{A}'$ is strictly diagonally dominant implying a unique solution $\mathbf{x}' = (x_2 \ \ldots \ x_{n-2})$ to the system $\mathbf{A}'\mathbf{x}' = \mathbf{c}'$. Using the obtained $x_2, x_{n-2}$ in (A.14),(A.15) we then get a unique solution $\mathbf{x} = (x_0 \ x_1 \ \ldots \ x_{n-1} \ x_n)$ to the system $\mathbf{A}\mathbf{x} = \mathbf{c}$ implying that $\mathbf{A}$ is invertible.

We now proceed to derive an upper bound on $\| \mathbf{A}^{-1} \|_\infty$. First note from Proposition 2 that $\| \mathbf{A}'^{-1} \|_\infty \leq 1/2$. Also note that $\| \mathbf{c}' \|_\infty \leq \| \mathbf{c} \|_\infty + 2(1/6) \| \mathbf{c} \|_\infty = (4/3) \| \mathbf{c} \|_\infty$. This then implies that $\| \mathbf{x}' \|_\infty \leq \| \mathbf{A}'^{-1} \|_\infty \| \mathbf{c}' \|_\infty \leq (2/3) \| \mathbf{c} \|_\infty$. From (A.14) we obtain $|x_1| \leq \| \mathbf{c} \|_\infty /3$ and also

$$|x_0| = |2x_1 - x_2 + c_0| \leq \frac{2 \| \mathbf{c} \|_\infty}{3} + \frac{2 \| \mathbf{c} \|_\infty}{3} + \| \mathbf{c} \|_\infty \leq \frac{7}{3} \| \mathbf{c} \|_\infty$$

where we used $|x_2| \leq \| \mathbf{x}' \|_\infty \leq (2/3) \| \mathbf{c} \|_\infty$. Similarly, from (A.15), we obtain $|x_{n-1}| \leq \| \mathbf{c} \|_\infty /3$ and $|x_n| \leq \frac{7}{3} \| \mathbf{c} \|_\infty$ repsectively. These bounds collectively imply $\| \mathbf{x} \|_\infty \leq (7/3) \| \mathbf{c} \|_\infty$. Lastly, we have that $\mathbf{x} = \mathbf{A}^{-1}\mathbf{c}$ implies

$$\| \mathbf{A}^{-1}\mathbf{c} \|_\infty = \| \mathbf{x} \|_\infty \leq \frac{7}{3} \| \mathbf{c} \|_\infty \quad \Rightarrow \quad \| \mathbf{A}^{-1} \frac{\mathbf{c}}{\| \mathbf{c} \|_\infty} \|_\infty \leq \frac{7}{3}.$$

Since the above holds for all $\mathbf{c}$ we obtain $\| \mathbf{A}^{-1} \|_\infty \leq (7/3)$. □

**Proposition 5.** *Consider the $(n-1) \times (n-1)$ matrix:*

$$
\mathbf{B} = \begin{pmatrix}
-2 & 1 & & & & \\
1 & -2 & 1 & & & \\
 & \ddots & \ddots & \ddots & & \\
 & & \ddots & \ddots & \ddots & \\
 & & & 1 & -2 & 1 \\
 & & & & 1 & -2
\end{pmatrix}.
$$

*We have that $\mathbf{B}$ is invertible and $\| \mathbf{B}^{-1} \|_\infty \leq \frac{n^2}{\pi^2}\sqrt{n-1}\left(1 - \frac{\pi^2}{12}\right)^{-1}$.*

*Proof.* It is well known (cf. "S. Noschese et al. Tridiagonal Toeplitz matrices: properties and novel applications, Numerical Linear Algebra with Applications, 2013" and references within) that eigenvalues $(\lambda_s)_{s=1}^{n-1}$ of $\mathbf{B}$ are given by $\lambda_s = -2 + 2\cos(s\pi/n)$. Clearly $\lambda_s \neq 0$ for $s = 1, \ldots, n-1$ implying $\mathbf{B}$ is invertible.

Next it is easy to see that $\| \mathbf{B}^{-1} \|_\infty \leq \| \mathbf{B}^{-1} \|_2 \sqrt{n-1}$. Indeed for any $\mathbf{x} \in \mathbb{R}^{(n-1)\times(n-1)}$:

$$\| \mathbf{B}^{-1}\mathbf{x} \|_\infty \leq \| \mathbf{B}^{-1}\mathbf{x} \|_2 \leq \| \mathbf{B}^{-1} \|_2 \| \mathbf{x} \|_\infty \sqrt{n-1}.$$

Since $\mathbf{B}$ is symmetric therefore:

$$\| \mathbf{B}^{-1} \|_\infty \leq \max_s \left| \lambda_s(\mathbf{B}^{-1}) \right| \sqrt{n-1} = \frac{1}{\min_s |\lambda_s(\mathbf{B})|} \sqrt{n-1}. \tag{A.16}$$

Now observe that

$$\min_s |\lambda_s(\mathbf{B})| = 2 \left( 1 - \cos\frac{\pi}{n} \right). \tag{A.17}$$

Using Taylors remainder theorem we obtain for some $\zeta \in (0, \pi/n)$:

$$\min_s |\lambda_s(\mathbf{B})| = 2 \left( 1 - \left( 1 - \frac{\pi^2}{2n^2} + \frac{\pi^4}{24n^4} \cos(\zeta) \right) \right), \tag{A.18}$$

$$= \frac{\pi^2}{n^2} - \frac{\pi^4}{12n^4} \cos(\zeta) \geq \frac{\pi^2}{n^2} \left( 1 - \frac{\pi^2}{12} \right). \tag{A.19}$$

$\square$

# B   Proof of Lemma 1

First recall, that we recover a stable approximation $\widehat{\mathbf{x}}_i$ to $\mathbf{x}_i$ via $\ell_1$ minimization [16, 17] as follows:

$$\widehat{\mathbf{x}}_i := \triangle(\mathbf{y}_i) := \operatorname*{argmin}_{\mathbf{y}_i = \mathbf{V}\mathbf{z}} \| \mathbf{z} \|_1. \tag{B.1}$$

Now in order to prove this lemma, we make use of a key theorem from [8]. While the first part is by now standard (see for example "R. Baraniuk et al., A Simple Proof of the Restricted Isometry Property for Random Matrices, Constructive Approximation, 2008"), the second result was stated in [8] as a specialization of Theorem 1.2 from [18] to the case of Bernoulli measurement matrices.

**Theorem 5** ([18, 8])**.** *Let $\mathbf{V}$ be a $m \times d$ random matrix with all entries being Bernoulli i.i.d random variables scaled with $1/\sqrt{m}$. Then the following results hold.*

1. *Let $0 < \mu < 1$. Then there are two positive constants $c_1, c_2 > 0$, such that the matrix $\mathbf{V}$ has the Restricted Isometry Property*

$$(1 - \mu) \| \mathbf{x} \|_2^2 \leq \| \mathbf{V}\mathbf{x} \|_2^2 \leq (1 + \mu) \| \mathbf{x} \|_2^2 \tag{B.2}$$

   *for all $\mathbf{x} \in \mathbb{R}^d$ such that $\#supp(\mathbf{x}) \leq c_2 m / \log(d/m)$ with probability at least $1 - e^{-c_1 m}$.*

2. *Let us suppose $d > (\log 6)^2 m$. Then there are positive constants $C, c_1', c_2' > 0$ such that with probability at least $1 - e^{-c_1' m} - e^{-\sqrt{md}}$ the matrix $\mathbf{V}$ has the following property. For every $\mathbf{x} \in \mathbb{R}^d$, $\mathbf{n} \in \mathbb{R}^m$ and every natural number $K \leq c_2' m / \log(d/m)$ we have*

$$\| \triangle(\mathbf{V}\mathbf{x} + \mathbf{n}) - \mathbf{x} \|_2 \leq C \left( K^{-1/2} \sigma_K(\mathbf{x})_1 + \max\left\{ \| \mathbf{n} \|_2, \sqrt{\log d} \| \mathbf{n} \|_\infty \right\} \right), \tag{B.3}$$

   *where*

$$\sigma_K(\mathbf{x})_1 := \inf \left\{ \| \mathbf{x} - \mathbf{z} \|_1 \colon \#supp(\mathbf{z}) \leq K \right\}$$

   *is the best $K$-term approximation of $\mathbf{x}$.*

**Remark 1.** *The proof of the second part of Theorem 5 requires* (B.2) *to hold, which is the case in our setting with high probability.*

Applying Theorem 5 to our setting we obtain the following corollary that bounds $\| \widehat{\mathbf{x}}_i - \mathbf{x}_i \|_2$ for all $i = -m_x, \ldots, m_x$ with high probability. This proves the first part of Lemma 1.

**Corollary 2.** *There exist constants $c_3' \geq 1$ and $C, c_1' > 0$ such that for $m_v$ satisfying $c_3' k \log d < m_v < d/(\log 6)^2$ we have with probability at least $1 - e^{-c_1' m_v} - e^{-\sqrt{m_v d}}$ that $\widehat{\mathbf{x}}_i$ as obtained in (B.1) satisfies for all $i = -m_x, \ldots, m_x$:*

$$\| \widehat{\mathbf{x}}_i - \mathbf{x}_i \|_2 \leq \frac{C \epsilon k B_2}{2 \sqrt{m_v}}, \tag{B.4}$$

*where $B_2 > 0$ is the constant defined in (2.1).*

*Proof.* We first note that $\mathbf{x}_i$ is at most $k$-sparse for each $i = -m_x, \ldots, m_x$ implying $\sigma_k(\mathbf{x}_i)_1 = 0$. This gives us:

$$\| \widehat{\mathbf{x}}_i - \mathbf{x}_i \|_2 \leq C \max \left\{ \| \mathbf{n}_i \|_2, \sqrt{\log d} \, \| \mathbf{n}_i \|_\infty \right\}; \ i = -m_x, \ldots, m_x. \tag{B.5}$$

Recall that $\mathbf{n}_i = [n_{i,1} \ldots n_{i,m_v}]$ where $n_{i,j} = \frac{1}{2} \epsilon \mathbf{v}_j^T \nabla^2 f(\zeta_{i,j}) \mathbf{v}_j$. Since $f(\zeta_{i,j}) = \sum_{l \in S} \phi_l(\zeta_{i,j}^{(l)})$ therefore $\nabla^2 f(\zeta_{i,j})$ is a diagonal matrix with the non-zero entries being $[\nabla^2 f(\zeta_{i,j})]_{l,l} = \phi_l''(\zeta_{i,j}^{(l)})$. Hence we have for all $i = -m_x, \ldots, m_x$ that

$$n_{i,j} = \frac{\epsilon}{2} \sum_{l \in S} v_{j,l} \phi_l''(\zeta_{i,j}^{(l)}) v_{j,l} \Rightarrow |n_{i,j}| \leq \frac{\epsilon}{2} \sum_{l \in S} \left| \phi_l''(\zeta_{i,j}^{(l)}) \right| v_{j,l}^2 \leq \frac{\epsilon k B_2}{2 m_v}$$

holds which then implies that $\| \mathbf{n}_i \|_2 = (\sum_{j=1}^{m_v} |n_{i,j}|^2)^{1/2} \leq \frac{\epsilon k B_2}{2 \sqrt{m_v}}$. Plugging this in (B.5) we get for the stated choice of $m_v$ that:

$$\| \widehat{\mathbf{x}}_i - \mathbf{x}_i \|_2 \leq C \max \left\{ \frac{\epsilon k B_2}{2 \sqrt{m_v}}, \sqrt{\log d} \frac{\epsilon k B_2}{2 m_v} \right\},$$

$$= \frac{C \epsilon k B_2}{2 \sqrt{m_v}} \max \left\{ 1, \frac{\sqrt{\log d}}{\sqrt{m_v}} \right\} = \frac{C \epsilon k B_2}{2 \sqrt{m_v}}.$$

$\square$

From Corollary 2 we observe that the decoding rule (B.1), for an appropriate choice of $m_v$, provides us with estimates: $\widehat{\phi'}_l(i/m_x)$ of $\phi_l'(i/m_x)$ for $i = -m_x, \ldots, m_x$ and for each $l = 1, \ldots, d$. We let $\tau = \frac{C \epsilon k B_2}{2 \sqrt{m_v}}$ denote the error bound of Corollary 2. Recall that we consider the following estimate to $S$:

$$\widehat{S} := \cup_{i=-m_x}^{m_x} \left\{ l \in \{1, \ldots, d\} : \left| \widehat{\phi'}_l(i/m_x) \right| > \tau \right\}. \tag{B.6}$$

The following Lemma gives precise conditions under which $\widehat{S} = S$ is guaranteed to hold. This proves the second part of Lemma 1.

**Lemma 4.** *Let $m_x$ be chosen such that $m_x \geq 1/\delta$ holds true with $\delta$ as defined in Assumption 3. Assuming that the conditions of Corollary 2 are satisfied, we have for any $\epsilon < \frac{D \sqrt{m_v}}{C k B_2}$ that $\widehat{S} = S$ holds true.*

*Proof.* Firstly observe that for all $i = -m_x, \ldots, m_x$ and $l = 1, \ldots, d$:

$$\| \widehat{\mathbf{x}}_i - \mathbf{x}_i \|_2 \leq \tau \ \Rightarrow \ \widehat{\phi'}_l(i/m_x) \in [\phi_l'(i/m_x) - \tau, \phi_l'(i/m_x) + \tau]. \tag{B.7}$$

Therefore for all $l \notin S$ we will have that $\widehat{\phi'}_l(i/m_x) \in [-\tau, \tau]$, $\forall i$. This means that we can recover all those $l \in S$ for which $\exists i \in \{-m_x, \ldots, m_x\}$ such that $\left| \widehat{\phi'}_l(i/m_x) \right| > \tau$ holds. Crucially, as a consequence of Assumption 3, we have for the choice $m_x \geq 1/\delta$, that for each $l \in S$, $\exists i \in \{-m_x, \ldots, m_x\}$ such that $|\phi_l'(i/m_x)| \geq D$ holds. Thus if $\tau < D/2$, then for all $l \in S$, $\left| \widehat{\phi'}_l(i/m_x) \right| > \tau$ for at least one $i$. Lastly, $\tau < D/2$ is ensured for the stated choice of $\epsilon$. $\square$

# C  Proof of Theorem 2

Before proceeding, we need to prove some secondary results that are used in the proof of Theorem 2. We begin in Section C.1 by first defining cubic splines formally and provide an approximation result for the not-a-knot cubic spline, in the noise-less setting. We note that this result is of independent interest. Next, in Section C.2 we provide an equivalent representation of (P) in terms of the coefficients of the cubic spline. Lastly, in Section C.3 we provide the proof of Theorem 2.

## C.1  Cubic splines and the not-a-knot cubic spline

We first begin by giving an overview of cubic spline interpolation in the setting where the exact values of the function (to be interpolated) are available. More formally let $t_1 = x_0 < x_1 < \cdots < x_{n-1} < x_n = t_2$ be a uniform mesh[9] where $x_{i+1} - x_i = h$. A spline of degree $p$ is a function $\mathcal{L}(x)$ which satisfies the following conditions:

1. For $x \in [x_i, x_{i+1}]$, $\mathcal{L}(x) = \mathcal{L}_i(x)$ (polynomial of degree at most $p$).
2. $\mathcal{L}^{(j)}$ exists and is continuous at the interior points $x_1, \ldots, x_{n-1}$ for all $1 \le j \le p - 1$.

Let $f \in C^4[t_1, t_2]$ be the function to be interpolated and say we are given $f_0, f_1, \ldots, f_n$ where $f_i = f(x_i)$. Our aim is to find a cubic spline : $\mathcal{L}$ such that $\mathcal{L}(x_i) = f_i$ for $i = 0, \ldots, n$. Hence in $[x_i, x_{i+1}]$, $\mathcal{L}_i(x)$ is a cubic polynomial such that the following conditions are satisfied.

1. $\mathcal{L}_i(x_i) = f_i$ and $\mathcal{L}_i(x_{i+1}) = f_{i+1}$ for $i = 0, \ldots, n - 1$. **(Interpolation condition)**
2. $\mathcal{L}'_i(x_{i+1}) = \mathcal{L}'_{i+1}(x_{i+1})$ for $i = 0, \ldots, n - 2$. **(Continuity of $\mathcal{L}'$ at $x_1, \ldots, x_{n-1}$)**
3. $\mathcal{L}''_i(x_{i+1}) = \mathcal{L}''_{i+1}(x_{i+1})$ for $i = 0, \ldots, n - 2$. **(Continuity of $\mathcal{L}''$ at $x_1, \ldots, x_{n-1}$)**

From (3) we have $\mathcal{L}''_i(x_i) = a_i = \mathcal{L}''_{i-1}(x_i)$. We can write $\mathcal{L}''_i$ as a Lagrange first order interpolating polynomial that interpolates $a_i$ and $a_{i+1}$ between $x_i$ and $x_{i+1}$ respectively. By integrating twice we have the following form for $\mathcal{L}_i(x)$:

$$\mathcal{L}_i(x) = \frac{a_i(x_{i+1} - x)^3}{6h} + \frac{a_{i+1}(x - x_i)^3}{6h} + b_i(x_{i+1} - x) + c_i(x - x_i). \tag{C.1}$$

Finding $\mathcal{L}$ is hence equivalent to finding the $(3n + 1)$ variables:

$$\mathbf{a} = (a_0, a_1, \ldots, a_n), \mathbf{b} = (b_0, \ldots, b_{n-1}), \mathbf{c} = (c_0, \ldots, c_{n-1}). \tag{C.2}$$

Note that Condition 1 involves $2n$ constraints while Condition 2 results in $n - 1$ constraints. Hence the above conditions give rise to $3n - 1$ equations leaving 2 free variables. There are several ways of taking care of this by introducing constraints at the boundary intervals (see [24] for examples).

**Not-a-knot cubic splines.**  One popular boundary condition was introduced in [24] and gives rise to the so-called "not-a-knot" cubic spline. This boundary condition does away with the requirement of having apriori knowledge about the derivatives of $f$ at the boundary points $x_0, x_n$. The idea here is to *not* treat $x_1, x_{n-1}$ as knots (hence the name) implying that there is a single cubic polynomial in the intervals $[x_0, x_2]$ and $[x_{n-2}, x_n]$ respectively. As each $\mathcal{L}_i$ is a cubic polynomial, this is equivalent to the boundary conditions:

$$\mathcal{L}'''_0(x_1) = \mathcal{L}'''_1(x_1) \text{ and } \mathcal{L}'''_{n-2}(x_{n-1}) = \mathcal{L}'''_{n-1}(x_{n-1}). \tag{C.3}$$

It is known that if $f \in C^4[t_1, t_2]$, then the approximation error is bounded[10] from above by $(40/64) \parallel f^{(4)} \parallel_\infty h^4$. It appears to the best of our knowledge, that the optimal error constant for this class of cubic splines is not known. We present in the form of the following proposition an improved error [11] bound of $(29/64) \parallel f^{(4)} \parallel_\infty h^4$ for uniform meshes. The proof technique is similar to that in "R.Beatson, On the Convergence of Some Cubic Spline Interpolation Schemes, SIAM Journal on Num. Anal.,1986". However the results there are for $C^2, C^3$ smooth $f$.

**Proposition 6.** *Consider $f \in C^4[t_1, t_2]$ and let $\mathcal{L}$ be the unique cubic spline obtained using the not-a-knot boundary conditions (C.3). We then have that:*

$$\| \mathcal{L} - f \|_\infty \leq \frac{29}{64} \| f^{(4)} \|_\infty h^4. \tag{C.4}$$

*Proof.* Recall from (C.1) that the cubic spline $\mathcal{L}$ has the following form in the interval $[x_i, x_{i+1}]$:

$$\mathcal{L}_i(x) = \frac{a_i(x_{i+1} - x)^3}{6h} + \frac{a_{i+1}(x - x_i)^3}{6h} + b_i(x_{i+1} - x) + c_i(x - x_i).$$

Now we have for any cubic spline the following well known identity (see [19] for instance):

$$\mathcal{L}''_{i-1} + 4\mathcal{L}''_i + \mathcal{L}''_{i+1} = \frac{6}{h^2}(\mathcal{L}_{i-1} - 2\mathcal{L}_i + \mathcal{L}_{i+1}); \quad 1 \leq i \leq n-1. \tag{C.5}$$

Recall the not-a-knot boundary conditions of (C.3):

$$\mathcal{L}'''_0(x_1) = \mathcal{L}'''_1(x_1) \quad \text{and} \quad \mathcal{L}'''_{n-2}(x_{n-1}) = \mathcal{L}'''_{n-1}(x_{n-1}).$$

Using (C.1) in (C.3) we equivalently obtain:

$$\mathcal{L}''_0 - 2\mathcal{L}''_1 + \mathcal{L}''_2 = 0 \quad \text{and} \quad \mathcal{L}''_{n-2} - 2\mathcal{L}''_{n-1} + \mathcal{L}''_n = 0. \tag{C.6}$$

(C.5) and (C.6) can then be written compactly as:

$$\underbrace{\begin{pmatrix} 1 & -2 & 1 & & & \\ 1 & 4 & 1 & & & \\ & 1 & 4 & 1 & & \\ & & \ddots & \ddots & \ddots & \\ & & & 1 & 4 & 1 \\ & & & & 1 & -2 & 1 \end{pmatrix}}_{\mathbf{A}} \begin{pmatrix} \mathcal{L}''_0 \\ \mathcal{L}''_1 \\ \vdots \\ \vdots \\ \mathcal{L}''_{n-1} \\ \mathcal{L}''_n \end{pmatrix} = \frac{6}{h^2} \begin{pmatrix} 0 \\ f_0 - 2f_1 + f_2 \\ \vdots \\ \vdots \\ f_{n-2} - 2f_{n-1} + f_n \\ 0 \end{pmatrix}. \tag{C.7}$$

For any smooth $f \in C^4[t_1, t_2]$ we will then have the following linear system:

$$\underbrace{\begin{pmatrix} 1 & -2 & 1 & & & \\ 1 & 4 & 1 & & & \\ & 1 & 4 & 1 & & \\ & & \ddots & \ddots & \ddots & \\ & & & 1 & 4 & 1 \\ & & & & 1 & -2 & 1 \end{pmatrix}}_{\mathbf{A}} \begin{pmatrix} f''_0 \\ f''_1 \\ \vdots \\ \vdots \\ f''_{n-1} \\ f''_n \end{pmatrix} = \frac{6}{h^2} \begin{pmatrix} 0 \\ f_0 - 2f_1 + f_2 \\ \vdots \\ \vdots \\ f_{n-2} - 2f_{n-1} + f_n \\ 0 \end{pmatrix} + \begin{pmatrix} R_0 \\ R_1 \\ \vdots \\ \vdots \\ R_{n-1} \\ R_n \end{pmatrix}. \tag{C.8}$$

Subtracting (C.7) from (C.8) it is easy to verify with the help of Proposition 4 and Proposition 3 that

$$\max_{0 \leq i \leq n} |\mathcal{L}''_i - f''_i| \leq \| \mathbf{A}^{-1} \|_\infty \max_{0 \leq i \leq n} |R_i| \leq \frac{7}{3} \left( \frac{3h^2 \| f^{(4)} \|_\infty}{2} \right) = \frac{7h^2 \| f^{(4)} \|_\infty}{2}. \tag{C.9}$$

We used Proposition 3(2) to obtain $|R_0|, |R_n| \leq h^2 \| f^{(4)} \|_\infty$. From Proposition 3(1) we have $|R_i| \leq \frac{3h^2 \| f^{(4)} \|_\infty}{2}$ for $1 \leq i \leq n-1$, which gives a uniform bound on $|R_i|$ for $0 \leq i \leq n$.

Lastly, Lemma 2 gives us the final error bound:

$$\| \mathcal{L} - f \|_\infty \leq \frac{h^4}{8} \left( \frac{7 \| f^{(4)} \|_\infty}{2} + \frac{1}{8} \| f^{(4)} \|_\infty \right) \leq \frac{29 \| f^{(4)} \|_\infty h^4}{64}.$$

$\square$

## C.2 An equivalent representation for (P)

The following lemma restates (P) in terms of the spline coefficients mentioned in (C.2).

**Lemma 5.** *We have the following representation of (P) in terms of the spline coefficients:* $\mathbf{a} = (a_0, a_1, \ldots, a_n), \mathbf{b} = (b_0, \ldots, b_{n-1}),$ *and* $\mathbf{c} = (c_0, \ldots, c_{n-1}).$

$$(P)\begin{cases}
\min \dfrac{h}{6}\mathbf{a}^T \begin{pmatrix} 2 & 1 & & & & \\ 1 & 4 & 1 & & & \\ & \ddots & \ddots & \ddots & & \\ & & \ddots & \ddots & \ddots & \\ & & & 1 & 4 & 1 \\ & & & & 1 & 2 \end{pmatrix} \mathbf{a} & \text{(C.10)} \\[2em]
\text{s.t.} \quad \widehat{g}_i - \gamma\tau \leq b_i h + \dfrac{a_i h^2}{6} \leq \widehat{g}_i + \gamma\tau; \quad i = 0, \ldots, n-1, & \text{(C.11)} \\[1em]
\widehat{g}_n - \gamma\tau \leq c_{n-1}h + \dfrac{a_n h^2}{6} \leq \widehat{g}_n + \gamma\tau, & \text{(C.12)} \\[1em]
c_i = b_{i+1}; \quad i = 0, \ldots, n-2, & \text{(C.13)} \\[0.5em]
a_{i+1}h = (b_i - c_i) + (c_{i+1} - b_{i+1}); \quad i = 0, \ldots, n-2, & \text{(C.14)} \\[0.5em]
a_0 - 2a_1 + a_2 = 0 \quad \text{and} \quad a_{n-2} - 2a_{n-1} + a_n = 0. & \text{(C.15)}
\end{cases}$$

*Proof.* Recall from (C.1) that $\mathcal{L}(x) = \mathcal{L}_i(x)$ for $x \in [x_i, x_{i+1}]$ where

$$\mathcal{L}_i(x) = \frac{a_i(x_{i+1} - x)^3}{6h} + \frac{a_{i+1}(x - x_i)^3}{6h} + b_i(x_{i+1} - x) + c_i(x - x_i).$$

Now we have that $\int_{t_1}^{t_2} \mathcal{L}''(x)^2 dx = \sum_{i=0}^{n-1} \int_{x_i}^{x_{i+1}} (\mathcal{L}_i''(x))^2 dx$. The following is easily verified.

$$\int_{x_i}^{x_{i+1}} (\mathcal{L}_i''(x))^2 dx = a_i^2 \frac{h}{3} + a_{i+1}^2 \frac{h}{3} + a_i a_{i+1} \frac{h}{3}. \tag{C.16}$$

This results in

$$\int_{t_1}^{t_2} \mathcal{L}''(x)^2 dx = \sum_{i=0}^{n-1} \left( a_i^2 \frac{h}{3} + a_{i+1}^2 \frac{h}{3} + a_i a_{i+1} \frac{h}{3} \right) \tag{C.17}$$

$$= \frac{h}{3} \left( a_0^2 + a_n^2 + 2(a_1^2 + \cdots + a_{n-1}^2) + \sum_{i=0}^{n-1} a_i a_{i+1} \right) \tag{C.18}$$

$$= \frac{h}{6} \left( 2a_0^2 + 2a_n^2 + 4\sum_{i=1}^{n-1} a_i^2 + 2\sum_{i=0}^{n-1} a_i a_{i+1} \right) \tag{C.19}$$

$$= \frac{h}{6}\mathbf{a}^T \begin{pmatrix} 2 & 1 & & & & \\ 1 & 4 & 1 & & & \\ & \ddots & \ddots & \ddots & & \\ & & \ddots & \ddots & \ddots & \\ & & & 1 & 4 & 1 \\ & & & & 1 & 2 \end{pmatrix} \mathbf{a} \tag{C.20}$$

where $\mathbf{a} = (a_0 \; a_1 \; \cdots \; a_{n-1} \; a_n)$. Note that the above matrix is strictly diagonally dominant and is also in fact positive definite. We now derive the constraints of the problem:

1. $\widehat{\mathbf{g}}_i - \gamma\tau \leq \mathcal{L}(\mathbf{x}_i) \leq \widehat{\mathbf{g}}_i + \gamma\tau; \quad \mathbf{i} = \mathbf{0}, \ldots, \mathbf{n}.$

We first note that: $\mathcal{L}(x_i) = \mathcal{L}_i(x_i)$ for $i = 0, \ldots, n-1$ and $\mathcal{L}(x_n) = \mathcal{L}_{n-1}(x_n)$. Hence by using (C.1) we obtain:

$$\widehat{g}_i - \gamma\tau \leq \mathcal{L}_{(}x_i) \leq \widehat{g}_i + \gamma\tau \Leftrightarrow \widehat{g}_i - \gamma\tau \leq b_i h + \frac{a_i h^2}{6} \leq \widehat{g}_i + \gamma\tau, \; i = 0, \ldots, n-1, \tag{C.21}$$

$$\widehat{g}_n - \gamma\tau \leq \mathcal{L}_{n-1}(x_n) \leq \widehat{g}_n + \gamma\tau \Leftrightarrow \widehat{g}_n - \gamma\tau \leq c_{n-1} h + \frac{a_n h^2}{6} \leq \widehat{g}_n + \gamma\tau. \tag{C.22}$$

2. **Continuity of $\mathcal{L}, \mathcal{L}', \mathcal{L}''$ at $\mathbf{x_1, \ldots, x_{n-1}}$.**

   First note that:

$$\mathcal{L}(x_i^-) = \mathcal{L}(x_i^+); \quad i = 1, \ldots, n-1 \tag{C.23}$$
$$\Leftrightarrow \mathcal{L}_i(x_{i+1}) = \mathcal{L}_{i+1}(x_{i+1}); \quad i = 0, \ldots, n-2 \tag{C.24}$$
$$\Leftrightarrow a_{i+1}\frac{h^2}{6} + c_i h = a_{i+1}\frac{h^2}{6} + b_{i+1}h; \quad i = 0, \ldots, n-2 \tag{C.25}$$
$$\Leftrightarrow c_i h = b_{i+1}h; \quad i = 0, \ldots, n-2. \tag{C.26}$$

   Next note that:

$$\mathcal{L}'(x_i^-) = \mathcal{L}'(x_i^+); \quad i = 1, \ldots, n-1 \tag{C.27}$$
$$\Leftrightarrow \mathcal{L}_i'(x_{i+1}) = \mathcal{L}_{i+1}'(x_{i+1}); \quad i = 0, \ldots, n-2 \tag{C.28}$$
$$\Leftrightarrow a_{i+1}\frac{h}{2} - b_i + c_i = -a_{i+1}\frac{h}{2} - b_{i+1} + c_{i+1}; \quad i = 0, \ldots, n-2 \tag{C.29}$$
$$\Leftrightarrow a_{i+1}h = (b_i - c_i) + (c_{i+1} - b_{i+1}); \quad i = 0, \ldots, n-2. \tag{C.30}$$

   The continuity of $\mathcal{L}''$ at $x_1, \ldots, x_{n-1}$ is already ensured through the choice of $a_i$'s.

3. **Continuity of $\mathcal{L}'''$ at $\mathbf{x_1, x_{n-1}}$ (not-a-knot boundary conditions).**

   It is easily verifiable that $\mathcal{L}_i'''(x) = -\frac{a_i}{h} + \frac{a_{i+1}}{h}$. Hence we have that

$$\mathcal{L}_0'''(x_1) = \mathcal{L}_1'''(x_1) \Leftrightarrow a_0 - 2a_1 + a_2 = 0, \tag{C.31}$$
$$\mathcal{L}_{n-2}'''(x_{n-1}) = \mathcal{L}_{n-1}'''(x_{n-1}) \Leftrightarrow a_{n-2} - 2a_{n-1} + a_n = 0. \tag{C.32}$$

$\square$

## C.3 Putting it together: Proof of Theorem 2

We prove a more general form of Theorem 2 namely the following.

**Theorem 6.** *For $g \in C^4[t_1, t_2]$ let $\mathcal{L}^* : [t_1, t_2] \to \mathbb{R}$ be a solution of (P) for some parameter $\gamma \geq 0$.*

1. *(General bound) For any $\gamma \geq 0$ we have that*

$$\| \mathcal{L}^* - g \|_\infty \leq \left[\frac{118(1 + \gamma)}{3}\right] \tau + \frac{29}{64}h^4 \| g^{(4)} \|_\infty . \tag{C.33}$$

2. *(Bound under large perturbation) For any $\gamma > 1$, if*

$$\tau \geq \frac{14h^2\lambda}{(\gamma - 1)} \left( \| g'' \|_\infty + \frac{h^2}{12} \| g^{(4)} \|_\infty \right) \tag{C.34}$$

*is satisfied where $\lambda = \Theta(n^{5/2})$ then $\mathcal{L}^*$ is a line and we have that*

$$\| \mathcal{L}^* - g \|_\infty \leq 2\tau \left[(\gamma + 1) + \frac{\gamma - 1}{3\lambda}\right] + \frac{29}{64}h^4 \| g^{(4)} \|_\infty . \tag{C.35}$$

*Proof.* **General error bound.** Let $(\mathbf{a}, \mathbf{b}, \mathbf{c})$ as defined in (C.2) be a feasible point satisfying (C.11)-(C.15). From (C.11),(C.12) we have that any feasible point satisfies:

$$b_i h = \widehat{g}_i - \frac{a_i h^2}{6} + e_i; \; i = 0, \ldots, n-1 \tag{C.36}$$

$$c_{n-1} h = \widehat{g}_n - \frac{a_n h^2}{6} + e_n, \tag{C.37}$$

for some $e_i$ with $|e_i| < \gamma\tau$. Plugging (C.36),(C.37), (C.13) in (C.14) we obtain for $i = 0, \ldots, n-2$:

$$a_{i+1} h = \frac{\widehat{g}_i - \frac{a_i h^2}{6} + e_i}{h} - 2\frac{\widehat{g}_{i+1} - \frac{a_{i+1} h^2}{6} + e_{i+1}}{h} + \frac{\widehat{g}_{i+2} - \frac{a_{i+2} h^2}{6} + e_{i+2}}{h}, \tag{C.38}$$

$$\Leftrightarrow a_{i+1} h = \left( \frac{\widehat{g}_i - 2\widehat{g}_{i+1} + \widehat{g}_{i+2}}{h} \right) + \left( \frac{a_{i+1} h}{3} - \frac{a_i h}{6} - \frac{a_{i+2} h}{6} \right) + \left( \frac{e_i - 2e_{i+1} + e_{i+2}}{h} \right), \tag{C.39}$$

$$\Leftrightarrow \frac{2a_{i+1} h}{3} + \frac{a_i h}{6} + \frac{a_{i+2} h}{6} = \left( \frac{\widehat{g}_i - 2\widehat{g}_{i+1} + \widehat{g}_{i+2}}{h} \right) + \left( \frac{e_i - 2e_{i+1} + e_{i+2}}{h} \right), \tag{C.40}$$

$$\Leftrightarrow a_i + 4_{i+1} + a_{i+2} = \frac{6}{h^2} (\widehat{g}_i - 2\widehat{g}_{i+1} + \widehat{g}_{i+2}) + \frac{6}{h^2} (e_i - 2e_{i+1} + e_{i+2}). \tag{C.41}$$

Now since $|e_i| \leq \gamma\tau$ we have that $\frac{6}{h^2} |(e_i - 2e_{i+1} + e_{i+2})| \leq \frac{24\gamma\tau}{h^2}$ for $i = 0, \ldots, n-2$. Furthermore, since $|\widehat{g}_i - g_i| \leq \tau$, for $i = 0, \ldots, n-2$ we have that

$$a_i + 4_{i+1} + a_{i+2} = \frac{6}{h^2} (g_i - 2g_{i+1} + g_{i+2}) + \eta_i; \quad i = 0, \ldots, n-2 \tag{C.42}$$

where $|\eta_i| \leq \frac{24(1+\gamma)\tau}{h^2}$. Thus (C.42) together with the boundary conditions (C.15) give us the following linear system of equations, perturbed by the noise vector $\eta$:

$$\underbrace{\begin{pmatrix} 1 & -2 & 1 & & & \\ 1 & 4 & 1 & & & \\ & 1 & 4 & 1 & & \\ & & \ddots & \ddots & \ddots & \\ & & & 1 & 4 & 1 \\ & & & 1 & -2 & 1 \end{pmatrix}}_{\mathbf{A}} \underbrace{\begin{pmatrix} a_0 \\ a_1 \\ \vdots \\ \vdots \\ a_{n-1} \\ a_n \end{pmatrix}}_{\mathbf{a}} = \frac{6}{h^2} \underbrace{\begin{pmatrix} 0 \\ g_0 - 2g_1 + g_2 \\ \vdots \\ \vdots \\ g_{n-2} - 2g_{n-1} + g_n \\ 0 \end{pmatrix}}_{\mathbf{g}} + \underbrace{\begin{pmatrix} 0 \\ \eta_0 \\ \vdots \\ \vdots \\ \eta_{n-2} \\ 0 \end{pmatrix}}_{\eta} \tag{C.43}$$

Now from Proposition 4 we know that $\| \mathbf{A}^{-1} \|_\infty \leq 7/3$. Also note that

$$\mathbf{a}_{or} := \mathbf{A}^{-1} \left( \frac{6}{h^2} \mathbf{g} \right) = (a_{0,or} \; a_{1,or} \; \cdots \; a_{n,or}) \tag{C.44}$$

denotes the (unique) coefficient vector that would have been obtained as solution in the noiseless setting. From (C.43) we then have that:

$$\mathbf{a} = \underbrace{\mathbf{A}^{-1} \left( \frac{6}{h^2} \mathbf{g} \right)}_{\mathbf{a}_{or}} + \mathbf{A}^{-1} \eta \tag{C.45}$$

$$\Rightarrow \| \mathbf{a} - \mathbf{a}_{or} \|_\infty \leq \| \mathbf{A}^{-1} \|_\infty \| \eta \|_\infty \leq \frac{56(1+\gamma)\tau}{h^2}. \tag{C.46}$$

Let $\mathbf{b}_{or} := (b_{0,or} \; b_{1,or} \; \cdots \; b_{n-1,or})$ and $\mathbf{c}_{or} := (c_{0,or} \; c_{1,or} \; \cdots \; c_{n-1,or})$ denote the optimal spline coefficients in the noise less setting ($\tau = 0$). Then, $b_{i,or} h = g_i - a_{i,or} \frac{h^2}{6}; \; i = 0, \ldots, n-1$. Furthermore $c_{i,or} = b_{i+1,or}; i = 0, \ldots, n-2$ and $c_{n-1,or} = g_n - a_{n,or} \frac{h^2}{6}$.

Now recall from (C.36) that $b_i h = \widehat{g}_i - a_i \frac{h^2}{6} + e_i$ where $|e_i| < \gamma\tau$. Since $b_{i,or} h = g_i - a_{i,or} \frac{h^2}{6}$, this then implies that

$$\| \mathbf{b} - \mathbf{b}_{or} \|_\infty = \max_{0 \le i \le n-1} |b_i - b_{i,or}|, \qquad (C.47)$$

$$\le \max_{0 \le i \le n-1} \frac{1}{h}\left( |\widehat{g}_i - g_i| + |e_i| + \frac{h^2}{6}|a_{i,or} - a_i| \right), \qquad (C.48)$$

$$\le \frac{1}{h}\left( (\gamma+1)\tau + \frac{28(1+\gamma)\tau}{3} \right) = \frac{31(1+\gamma)\tau}{3h}. \qquad (C.49)$$

Analogously, it can be verified that $\| \mathbf{c} - \mathbf{c}_{or} \|_\infty \le \frac{31(1+\gamma)\tau}{3h}$.

Now let $\mathcal{L}_{or}$ denote the (unique) cubic spline obtained in the noiseless setting ($\tau = 0$) so that $\mathcal{L}_{or}(x) = \mathcal{L}_{i,or}(x)$ for $x \in [x_i, x_{i+1}]$ where

$$\mathcal{L}_{or}(x) = \frac{a_{i,or}(x_{i+1} - x)^3}{6h} + \frac{a_{i+1,or}(x - x_i)^3}{6h} + b_{i,or}(x_{i+1} - x) + c_{i,or}(x - x_i). \qquad (C.50)$$

Let $\mathbf{a}^*, \mathbf{b}^*, \mathbf{c}^*$ be a solution to (P) and $\mathcal{L}^*$ denote the corresponding cubic spline. We then have for $x \in [x_i, x_{i+1}]$ that :

$$|\mathcal{L}_i^*(x) - \mathcal{L}_{i,or}(x)| = \left| (a_i^* - a_{i,or})\frac{(x_{i+1} - x)^3}{6h} + (a_{i+1}^* - a_{i+1,or})\frac{(x - x_i)^3}{6h} \right. \qquad (C.51)$$

$$+ \left. (b_i^* - b_{i,or})(x_{i+1} - x) + (c_i^* - c_{i,or})(x - x_i) \right|, \qquad (C.52)$$

$$\le \| \mathbf{a}^* - \mathbf{a}_{or} \|_\infty \left| \frac{(x_{i+1} - x)^3}{6h} \right| + \| \mathbf{a}^* - \mathbf{a}_{or} \|_\infty \left| \frac{(x - x_i)^3}{6h} \right| \qquad (C.53)$$

$$+ \| \mathbf{b}^* - \mathbf{b}_{or} \|_\infty |x_{i+1} - x| + \| \mathbf{c}^* - \mathbf{c}_{or} \|_\infty |x - x_i|, \qquad (C.54)$$

$$\le \frac{56(1+\gamma)\tau}{3} + \frac{62(1+\gamma)\tau}{3} = \frac{118(1+\gamma)\tau}{3}. \qquad (C.55)$$

This then implies that

$$\| \mathcal{L}^* - \mathcal{L}_{or} \|_\infty = \max_{0 \le i \le n-1} \| \mathcal{L}_i^* - \mathcal{L}_{i,or} \|_\infty \le \frac{118(1+\gamma)\tau}{3}. \qquad (C.56)$$

Since $\| \mathcal{L}_{or} - g \|_\infty \le \frac{29}{64}h^4 \| g^{(4)} \|_\infty$ by Proposition 6, we have by triangles inequality the final error bound:

$$\| \mathcal{L}^* - g \|_\infty \le \| \mathcal{L}^* - \mathcal{L}_{or} \|_\infty + \| \mathcal{L}_{or} - g \|_\infty, \qquad (C.57)$$

$$\le \frac{118(1+\gamma)\tau}{3} + \frac{29}{64}h^4 \| g^{(4)} \|_\infty. \qquad (C.58)$$

**Bound under large perturbation.** Let $\mathbf{a}_{or}, \mathbf{b}_{or}, \mathbf{c}_{or}$ denote the unique solution when $\tau = 0$, i.e. in the noiseless setting. Recall that $\mathbf{a}_{or}$ has the form:

$$\mathbf{a}_{or} = \mathbf{A}^{-1}\left( \frac{6}{h^2} \right) \begin{pmatrix} 0 \\ g_0 - 2g_1 + g_2 \\ \vdots \\ \vdots \\ g_{n-2} - 2g_{n-1} + g_n \\ 0 \end{pmatrix}. \qquad (C.59)$$

Using Proposition 3(1) we then obtain:

$$\| \mathbf{a}_{or} \|_\infty \le 6 \| \mathbf{A}^{-1} \|_\infty \max_{1 \le i \le n-1}\left( \left| g_i'' + \frac{h^2}{12} \| g^{(4)} \|_\infty \right| \right),$$

$$\le 14\left( \| g'' \|_\infty + \frac{h^2}{12} \| g^{(4)} \|_\infty \right), \qquad (C.60)$$

where we used $\| \mathbf{A}^{-1} \|_\infty \leq 7/3$ from Proposition 4. Recall also that:

$$b_{i,or} = \frac{g_i}{h} - \frac{a_{i,or}h}{6}; \quad i = 0, \ldots, n-1, \tag{C.61}$$

$$c_{i,or} = b_{i+1,or}; \quad i = 0, \ldots, n-2, \tag{C.62}$$

$$\text{and} \quad c_{n-1,or} = \frac{g_n}{h} - \frac{a_{n,or}h}{6}. \tag{C.63}$$

Let $(\mathbf{a}^*, \mathbf{b}^*, \mathbf{c}^*)$ be a solution to (P). Then $\mathbf{a}^* = \mathbf{0}$ iff there exists feasible $\mathbf{b}^*, \mathbf{c}^*$ satisfying:

$$b_i^* \in \left[ \frac{\widehat{g}_i - \gamma\tau}{h}, \frac{\widehat{g}_i + \gamma\tau}{h} \right]; \quad i = 0, \ldots, n-1, \tag{C.64}$$

$$\begin{pmatrix} -2 & 1 & & & & \\ 1 & -2 & 1 & & & \\ & \ddots & \ddots & \ddots & & \\ & & \ddots & \ddots & \ddots & \\ & & & 1 & -2 & 1 \\ & & & & 1 & -2 \end{pmatrix} \begin{pmatrix} b_1^* \\ b_2^* \\ \vdots \\ \vdots \\ b_{n-2}^* \\ b_{n-1}^* \end{pmatrix} = \begin{pmatrix} -b_0^* \\ 0 \\ \vdots \\ \vdots \\ 0 \\ -c_{n-1}^* \end{pmatrix}, \tag{C.65}$$

$$c_{n-1}^* \in \left[ \frac{\widehat{g}_n - \gamma\tau}{h}, \frac{\widehat{g}_n + \gamma\tau}{h} \right]. \tag{C.66}$$

Here (C.65) follows by plugging (C.13) in (C.14) and using $\mathbf{a}^* = \mathbf{0}$. Now we know that $\mathbf{b}_{or}, \mathbf{c}_{or}$ satisfy:

$$\underbrace{\begin{pmatrix} -2 & 1 & & & & \\ 1 & -2 & 1 & & & \\ & \ddots & \ddots & \ddots & & \\ & & \ddots & \ddots & \ddots & \\ & & & 1 & -2 & 1 \\ & & & & 1 & -2 \end{pmatrix}}_{\mathbf{B}} \begin{pmatrix} b_{1,or} \\ b_{2,or} \\ \vdots \\ \vdots \\ b_{n-2,or} \\ b_{n-1,or} \end{pmatrix} = \begin{pmatrix} -b_{0,or} \\ 0 \\ \vdots \\ \vdots \\ 0 \\ -c_{n-1,or} \end{pmatrix} + h \begin{pmatrix} a_{1,or} \\ a_{2,or} \\ \vdots \\ \vdots \\ a_{n-2,or} \\ a_{n-1,or} \end{pmatrix}. \tag{C.67}$$

This follows simply from the constraints (C.13),(C.14). Since from Proposition 5 we have that $\mathbf{B}$ is invertible, hence we can rewrite (C.67) to obtain equivalently:

$$\mathbf{B} \left[ \begin{pmatrix} b_{1,or} \\ b_{2,or} \\ \vdots \\ \vdots \\ b_{n-2,or} \\ b_{n-1,or} \end{pmatrix} - h\mathbf{B}^{-1} \underbrace{\begin{pmatrix} a_{1,or} \\ a_{2,or} \\ \vdots \\ \vdots \\ a_{n-2,or} \\ a_{n-1,or} \end{pmatrix}}_{\mathbf{a}'} \right] = \begin{pmatrix} -b_{0,or} \\ 0 \\ \vdots \\ \vdots \\ 0 \\ -c_{n-1,or} \end{pmatrix}. \tag{C.68}$$

Observe that (C.68) holds for $\mathbf{a}_{or}, \mathbf{b}_{or}, \mathbf{c}_{or}$. Thus if we can find conditions on $\tau$ such that

$$\frac{\widehat{g}_0 - \gamma\tau}{h} \leq \underbrace{\frac{g_0}{h} - \frac{a_{0,or}h}{6}}_{b_{0,or}} \leq \frac{\widehat{g}_0 + \gamma\tau}{h}, \tag{C.69}$$

$$\frac{\widehat{g}_i - \gamma\tau}{h} \leq \underbrace{\left( \frac{g_i}{h} - \frac{a_{i,or}h}{6} \right)}_{b_{i,or}} - h(\mathbf{B}^{-1}\mathbf{a}')_i \leq \frac{\widehat{g}_i + \gamma\tau}{h}; \quad i = 1, \ldots, n-1, \tag{C.70}$$

$$\frac{\widehat{g}_n - \gamma\tau}{h} \leq \underbrace{\frac{g_n}{h} - \frac{a_{n,or}h}{6}}_{c_{n-1,or}} \leq \frac{\widehat{g}_n + \gamma\tau}{h} \tag{C.71}$$

are satisfied then we are done. Let us look at (C.70) first. We equivalently obtain for each $i = 1, \ldots, n-1$:

$$\frac{\gamma\tau}{h} \geq \frac{g_i - \widehat{g}_i}{h} - \frac{a_{i,or}h}{6} - h(\mathbf{B}^{-1}\mathbf{a}')_i, \tag{C.72}$$

$$\frac{\gamma\tau}{h} \geq \frac{\widehat{g}_i - g_i}{h} + \frac{a_{i,or}h}{6} + h(\mathbf{B}^{-1}\mathbf{a}')_i. \tag{C.73}$$

Clearly, R.H.S of (C.72),(C.73) is less than or equal to: $\frac{\tau}{h} + \frac{\|\mathbf{a}_{or}\|_\infty h}{6} + h \parallel \mathbf{B}^{-1} \parallel_\infty \parallel \mathbf{a}_{or} \parallel_\infty$. Thus a sufficient condition for (C.72),(C.73) to hold is:

$$\frac{\gamma\tau}{h} \geq \frac{\tau}{h} + \frac{\parallel \mathbf{a}_{or} \parallel_\infty h}{6} + h \parallel \mathbf{B}^{-1} \parallel_\infty \parallel \mathbf{a}_{or} \parallel_\infty, \tag{C.74}$$

$$\Leftrightarrow \tau \geq \frac{h^2}{\gamma - 1} \parallel \mathbf{a}_{or} \parallel_\infty \left( \frac{1}{6} + \parallel \mathbf{B}^{-1} \parallel_\infty \right). \tag{C.75}$$

Using (C.60) and Proposition 5 the sufficient condition becomes:

$$\tau \geq \frac{14h^2}{\gamma - 1} \left( \parallel g'' \parallel_\infty + \frac{h^2}{12} \parallel g^{(4)} \parallel_\infty \right) \left( \frac{1}{6} + \frac{n^2}{\pi^2}\sqrt{n-1}\left(1 - \frac{\pi^2}{12}\right)^{-1} \right). \tag{C.76}$$

Observe that the above condition on $\tau$ also guarantees that (C.69), (C.71) are satisfied. Hence if the error $\tau$ is such that it satisfies (C.76) we then have that $\mathbf{a}^* = 0$. Using (C.60), this then implies that:

$$\parallel \mathbf{a}_{or} - \mathbf{a}^* \parallel_\infty \leq 14 \left( \parallel g'' \parallel_\infty + \frac{h^2}{12} \parallel g^{(4)} \parallel_\infty \right). \tag{C.77}$$

Furthermore since

$$b_{i,or} = \frac{g_i}{h} - \frac{a_{i,or}h}{6}, \text{ and } b_i^* \in \left[ \frac{\widehat{g}_i - \gamma\tau}{h}, \frac{\widehat{g}_i + \gamma\tau}{h} \right]; \quad i = 0, \ldots, n-1 \tag{C.78}$$

holds, we have that:

$$\parallel \mathbf{b}_{or} - \mathbf{b}^* \parallel_\infty \leq \frac{\tau}{h} + \frac{\gamma\tau}{h} + \parallel \mathbf{a}_{or} \parallel_\infty \frac{h}{6}, \tag{C.79}$$

$$\leq \frac{\tau(\gamma + 1)}{h} + \frac{7h}{3} \left( \parallel g'' \parallel_\infty + \frac{h^2}{12} \parallel g^{(4)} \parallel_\infty \right). \tag{C.80}$$

Analogously we obtain

$$\parallel \mathbf{c}_{or} - \mathbf{c}^* \parallel_\infty \leq \frac{\tau(\gamma + 1)}{h} + \frac{7h}{3} \left( \parallel g'' \parallel_\infty + \frac{h^2}{12} \parallel g^{(4)} \parallel_\infty \right). \tag{C.81}$$

Let $\mathcal{L}_{or}, \mathcal{L}^*$ be the cubic splines corresponding to $(\mathbf{a}_{or}, \mathbf{b}_{or}, \mathbf{c}_{or})$ and $(\mathbf{a}^*(= \mathbf{0}), \mathbf{b}^*, \mathbf{c}^*)$. We then have for $x \in [x_i, x_{i+1}]$:

$$
\begin{aligned}
|\mathcal{L}_i^*(x) - \mathcal{L}_{i,or}(x)| &= |(a_i^* - a_{i,or})\frac{(x_{i+1} - x)^3}{6h} + (a_{i+1}^* - a_{i+1,or})\frac{(x - x_i)^3}{6h} && \text{(C.82)} \\
&+ (b_i^* - b_{i,or})(x_{i+1} - x) + (c_i^* - c_{i,or})(x - x_i)|, && \text{(C.83)} \\
&\leq \parallel \mathbf{a}^* - \mathbf{a}_{or} \parallel_\infty \left|\frac{(x_{i+1} - x)^3}{6h}\right| + \parallel \mathbf{a}^* - \mathbf{a}_{or} \parallel_\infty \left|\frac{(x - x_i)^3}{6h}\right| && \text{(C.84)} \\
&+ \parallel \mathbf{b}^* - \mathbf{b}_{or} \parallel_\infty |x_{i+1} - x| + \parallel \mathbf{c}^* - \mathbf{c}_{or} \parallel_\infty |x - x_i|, && \text{(C.85)} \\
&\leq \parallel \mathbf{a}^* - \mathbf{a}_{or} \parallel_\infty \frac{h^2}{3} + 2h \parallel \mathbf{b}^* - \mathbf{b}_{or} \parallel_\infty, && \text{(C.86)} \\
&\leq 2(\gamma + 1)\tau + \frac{28h^2}{3} \left( \parallel g'' \parallel_\infty + \frac{h^2}{12} \parallel g^{(4)} \parallel_\infty \right). && \text{(C.87)}
\end{aligned}
$$

This in turn implies that

$$\parallel \mathcal{L}^* - \mathcal{L}_{or} \parallel_\infty = \max_{0 \leq i \leq n-1} \parallel \mathcal{L}_i^* - \mathcal{L}_{i,or} \parallel_\infty \leq 2(\gamma + 1)\tau + \frac{28h^2}{3} \left( \parallel g'' \parallel_\infty + \frac{h^2}{12} \parallel g^{(4)} \parallel_\infty \right). \tag{C.88}$$

Since by Proposition 6: $\parallel \mathcal{L}_{or} - g \parallel_\infty \leq \frac{29}{64}h^4 \parallel g^{(4)} \parallel_\infty$, we finally obtain the stated error bound by employing (C.76) in (C.88), and then by using triangles inequality. $\qquad\square$

## C.4 Uniqueness of the solution of (P)

In this section we will discuss about the uniqueness of the solution returned by the convex program (P). We state this in the form of the following theorem.

**Theorem 7.** *Let $\mathcal{L}^* : [t_1, t_2] \to \mathbb{R}$ be one of the optimal solutions to (P), for any fixed $\gamma \geq 0$. If $\int_{t_1}^{t_2} (\mathcal{L}^{*\prime\prime}(x))^2 dx > 0$, then $\mathcal{L}^*$ is a unique solution.*

*Proof.* Let $\mathbf{a}^* = (a_0^*, \dots, a_n^*)$, $\mathbf{b}^* = (b_0^*, \dots, b_{n-1}^*)$, $\mathbf{c}^* = (c_0^*, \dots, c_{n-1}^*)$ be a solution to (P), that corresponds to the spline $\mathcal{L}^*$. This implies that $\mathbf{a}^*, \mathbf{b}^*, \mathbf{c}^*$ satisfy:

$$\widehat{g}_i - \gamma\tau \leq b_i h + \frac{a_i h^2}{6} \leq \widehat{g}_i + \gamma\tau; \quad i = 0, \dots, n-1, \tag{C.89}$$

$$\widehat{g}_n - \gamma\tau \leq c_{n-1} h + \frac{a_n h^2}{6} \leq \widehat{g}_n + \gamma\tau, \tag{C.90}$$

$$\underbrace{\begin{pmatrix} -2 & 1 & & & & \\ 1 & -2 & 1 & & & \\ & \ddots & \ddots & \ddots & & \\ & & \ddots & \ddots & \ddots & \\ & & & 1 & -2 & 1 \\ & & & & 1 & -2 \end{pmatrix}}_{\mathbf{B}} \begin{pmatrix} b_1 \\ b_2 \\ \vdots \\ \vdots \\ b_{n-2} \\ b_{n-1} \end{pmatrix} = \begin{pmatrix} -b_0 \\ 0 \\ \vdots \\ \vdots \\ 0 \\ -c_{n-1} \end{pmatrix} + h \begin{pmatrix} a_1 \\ a_2 \\ \vdots \\ \vdots \\ a_{n-2} \\ a_{n-1} \end{pmatrix}, \tag{C.91}$$

$$a_0 - 2a_1 + a_2 = 0 \quad \text{and} \quad a_{n-2} - 2a_{n-1} + a_n = 0. \tag{C.92}$$

Here, (C.91) is derived from (C.13), (C.14) leading to the variables $c_0, \dots, c_{n-2}$ being eliminated. We also have that $\int_{t_1}^{t_2}(\mathcal{L}^{*\prime\prime}(x))^2 dx = \frac{h}{6}\mathbf{a}^{*T}Q\mathbf{a}^*$, where $Q$ is the (positive definite) matrix in (C.10). Therefore, since $\int_{t_1}^{t_2}(\mathcal{L}^{*\prime\prime}(x))^2 dx > 0$ we must have at least one non-zero $a_i^*$.

Let us denote the $i^{th}$ constraint in (C.89) by $E_i$ and the constraint in (C.90) by $E_n$.

**Claim 1.** *Not all the $E_i$; $i = 0, \dots, n$ are satisfied strictly by $\mathbf{a}^*, \mathbf{b}^*, c_{n-1}^*$. In other words we cannot have that*

$$\widehat{g}_i - \gamma\tau < b_i^* h + \frac{a_i^* h^2}{6} < \widehat{g}_i + \gamma\tau; \quad i = 0, \dots, n-1,$$

$$\widehat{g}_n - \gamma\tau < c_{n-1}^* h + \frac{a_n^* h^2}{6} < \widehat{g}_n + \gamma\tau.$$

*Proof.* Let us assume the contrary. Then this implies that $\exists t \in (0, 1)$, such that $(t\mathbf{a}^*, t\mathbf{b}^*, tc_{n-1}^*)$ is also feasible. However $\int_{t_1}^{t_2}(\mathcal{L}^{*\prime\prime}(x))^2 dx$ has a smaller value at $t\mathbf{a}^*$ than $\mathbf{a}^*$, which is a contradiction to $\mathbf{a}^*$ being optimal. $\square$

Now say that $\mathbf{a}' = (a_0', \dots, a_n')$, $\mathbf{b}' = (b_0', \dots, b_{n-1}')$, $c_{n-1}'$ be another solution to (P) where: $\mathbf{a}' \neq \mathbf{a}^*$ and $\mathbf{a}^{*T}Q\mathbf{a}^* = \mathbf{a}'^{T}Q\mathbf{a}'$. Clearly, any $\tilde{\mathbf{a}}, \tilde{\mathbf{b}}, \tilde{c}_{n-1}$ with: $\tilde{\mathbf{a}} = \lambda\mathbf{a}^* + (1-\lambda)\mathbf{a}'$, $\tilde{\mathbf{b}} = \lambda\mathbf{b}^* + (1-\lambda)\mathbf{b}'$, $\tilde{c}_{n-1} = \lambda c_{n-1}^* + (1-\lambda)c_{n-1}'$ for $\lambda \in (0, 1)$ is also feasible. However, it can be verified that for $\lambda \in (0, 1)$, we have $\tilde{\mathbf{a}}^T Q\tilde{\mathbf{a}} < \mathbf{a}^{*T}Q\mathbf{a}^*$ which is a contradiction.

Let $\mathbf{a}^*, \mathbf{b}' = (b_0', \dots, b_{n-1}'), c_{n-1}'$ be another solution to (P), corresponding to a spline $\mathcal{L}_1$, with $\mathcal{L}_1 \neq \mathcal{L}^*$. This means that *at least* one of $b_i^*, c_{n-1}^*$ differs from $b_i', c_{n-1}'$; $i = 0, \dots, n-1$.

**Claim 2.** *It is necessary that: $b_0^* \neq b_0'$ and/or $c_{n-1}^* \neq c_{n-1}'$*

*Proof.* In case both $b_0^* = b_0'$ and $c_{n-1}^* = c_{n-1}'$ holds, then due to invertibility of $\mathbf{B}$, we have $b_i^* = b_i'$; $i = 0, \dots, n-1$. This means that $\mathcal{L}_1 = \mathcal{L}^*$, contradicting our assumption that they are different. $\square$

**Lemma 6.** *If $b_0^* \neq b_0'$ and/or $c_{n-1}^* \neq c_{n-1}'$, then it implies that the solution $\mathbf{a}^*, \mathbf{b}^*, c_{n-1}^*$ is unique.*

*Proof.* On account of Claim 2, we first consider the scenario where $b'_0 = b^*_0 + \epsilon$ for some $\epsilon > 0$. For $\epsilon_1, \epsilon_2 \geq 0$ we then have one of the following four possibilities for $\mathbf{b}' = (b'_0, \ldots, b'_{n-1}), c'_{n-1}$.

1. **Case 1.** $b'_1 = b^*_1 + \epsilon_1$ and $b'_2 = b^*_2 + \epsilon_2$. Due to (C.91), this then implies that $2\epsilon_1 - \epsilon_2 = \epsilon$ and

$$b'_i = b^*_i - [(i-2)\epsilon_1 - (i-1)\epsilon_2]; \quad i = 3, \ldots, n-1,$$
$$c'_{n-1} = c^*_{n-1} - [(n-2)\epsilon_1 - (n-1)\epsilon_2].$$

2. **Case 2.** $b'_1 = b^*_1 - \epsilon_1$ and $b'_2 = b^*_2 + \epsilon_2$. Due to (C.91), this then implies that $2\epsilon_1 + \epsilon_2 = -\epsilon$ which is impossible. Hence this case cannot occur.

3. **Case 3.** $b'_1 = b^*_1 + \epsilon_1$ and $b'_2 = b^*_2 - \epsilon_2$, which on account of (C.91) implies that $2\epsilon_1 + \epsilon_2 = \epsilon$ and

$$b'_i = b^*_i - [(i-2)\epsilon_1 + (i-1)\epsilon_2]; \quad i = 3, \ldots, n-1,$$
$$c'_{n-1} = c^*_{n-1} - [(n-2)\epsilon_1 + (n-1)\epsilon_2].$$

4. **Case 4.** $b'_1 = b^*_1 - \epsilon_1$ and $b'_2 = b^*_2 - \epsilon_2$. On account of (C.91), this then implies that $\epsilon_2 - 2\epsilon_1 = \epsilon$ and

$$b'_i = b^*_i - [(i-1)\epsilon_2 - (i-2)\epsilon_1]; \quad i = 3, \ldots, n-1,$$
$$c'_{n-1} = c^*_{n-1} - [(n-1)\epsilon_2 - (n-2)\epsilon_1].$$

Say that it is possible to obtain $\mathbf{b}', c'_{n-1}$ in one of the above cases, for some value of $\epsilon > 0$ and $\epsilon_1, \epsilon_2 \geq 0$. We then make the following claim.

**Claim 3.** *There will always exist $\epsilon', \epsilon'_1, \epsilon'_2 > 0$ resulting in $\mathbf{b}', c'_{n-1}$ such that each $E_i; i = 0, \ldots, n$ is satisfied strictly by $\mathbf{a}^*, \mathbf{b}', c'_{n-1}$.*

*Proof.* Consider Case 1 first. Denote $P = \left\{ 0, \frac{1}{2}, \frac{2}{3}, \ldots, \frac{n-2}{n-1} \right\}$. We necessarily have $\epsilon_1 > 0$, however $\epsilon_2 \geq 0$. Let $\epsilon_2 = \ell\epsilon_1$, where $0 \leq \ell < 2$. This implies that

$$\epsilon_1 = \frac{\epsilon}{2-\ell}, \quad \epsilon_2 = \frac{\ell\epsilon}{2-\ell},$$
$$b'_i = b^*_i - \frac{\epsilon}{2-\ell}[(i-2) - (i-1)\ell]; \quad i = 3, \ldots, n-1,$$
$$c'_{n-1} = c^*_{n-1} - \frac{\epsilon}{2-\ell}[(n-2) - (n-1)\ell].$$

If $\ell \in P$ then at most one $b^*_i$ or $c^*_{n-1}$ is equal to the corresponding $b'_i$ or $c'_{n-1}$, repsectively. For any other $\ell$, we will have that $b^*_i \neq b'_i$ and $c^*_{n-1} \neq c'_{n-1}; i = 0, \ldots, n-1$. Note that if $\ell \notin P$, then $\exists \epsilon' < \epsilon$ such that with $\ell, \epsilon'$, we have that each $b^*_i$ and $c^*_{n-1}$ changes by a non-zero amount with $b'_i, c'_{n-1}$ lying strictly inside the intervals defined in (C.89),(C.90). In case $\ell \in P$, we have for $\ell' = \beta + \ell$, with $\beta \neq 0$ chosen appropriately, that $\ell' \notin P$. For this $\ell'$ we can then choose $\epsilon' < \epsilon$ suitably to ensure that each $b^*_i$ and $c^*_{n-1}$ changes by a non-zero amount with $b'_i, c'_{n-1}$ strictly satisfying (C.89),(C.90). Similar arguments can be made for Case 3 and Case 4. $\qquad\square$

Consequently, with the help of Claim 1, we arrive at a contradiction regarding optimality of $\mathbf{a}^*$. This implies that none of Cases $1-4$ are possible. In the scenario where $b'_0 = b^*_0 - \epsilon$ for some $\epsilon > 0$, we arrive via an analogous case analysis at the same conclusion as above.

Lastly, we could have also started the above analysis by considering $c'_{n-1} = c^*_{n-1} \pm \epsilon$, for $\epsilon > 0$. Indeed, by considering different possibilities for perturbation of $b^*_{n-1}, b^*_{n-2}$, we will obtain perturbation expressions for all: $b'_{n-3}, b'_{n-4}, \ldots, b'_1, b'_0$. By an analogous case analysis as shown above, we can again show that none of the cases are possible. This means that $\nexists \mathbf{a}^*, \mathbf{b}', c'_{n-1}$ different from $\mathbf{a}^*, \mathbf{b}^*, c^*_{n-1}$, as a solution to (P).

$\qquad\square$

$\qquad\square$

# D Proof of Theorem 1

In this section, we will prove a general version of Theorem 1 namely the following.

**Theorem 8.** *There exist constants $C, C_1 > 0$ such that if $m_x \geq (1/\delta)$, $m_v \geq C_1 k \log d$, $0 < \epsilon < \frac{D\sqrt{m_v}}{CkB_2}$ and $\tau = \frac{C\epsilon kB_2}{2\sqrt{m_v}}$ then with high probability, $\widehat{S} = S$ and for any $\gamma \geq 0$ the estimate $\phi_{est,l}$ returned by Algorithm 1 satisfies for each $l \in S$:*

$$\| \phi_{est,l} - \phi_l \|_{L^\infty[-1,1]} \leq [59(1+\gamma)] \frac{C\epsilon kB_2}{\sqrt{m_v}} + \frac{87}{64m_x^4} \| \phi_l^{(5)} \|_{L^\infty[-1,1]} . \tag{D.1}$$

*Furthermore for any $\gamma > 1$ suppose that $\epsilon$ additionally satisfies*

$$\epsilon > C_3\sqrt{m_v m_x} \left( \| \phi_l''' \|_{L^\infty[-1,1]} + \frac{1}{12m_x^2} \| \phi_l^{(5)} \|_{L^\infty[-1,1]} \right) \tag{D.2}$$

*for some constant $C_3$ that depends on $C, k, B_2, \gamma$. Then the estimate $\phi_{est,l}$ is a polynomial of degree at most 2 and:*

$$\| \phi_{est,l} - \phi_l \|_{L^\infty[-1,1]} \leq \left[ 3(\gamma+1) + \frac{\gamma-1}{\lambda} \right] \frac{C\epsilon kB_2}{\sqrt{m_v}} + \frac{87}{64m_x^4} \| \phi_l^{(5)} \|_{L^\infty[-1,1]} . \tag{D.3}$$

To this end we first have the following Corollary of Theorem 6 for estimation of $C^4$ smooth $\phi_l'$ in the interval $[-1, 1]$. Corollary 3 can be seen as a generalized version of Corollary 1.

**Corollary 3.** *Let (P) be employed for each $l \in S$ using noisy samples $\left\{ \widehat{\phi}'_l(i/m_x) \right\}_{i=-m_x}^{m_x}$, and with step size $\epsilon$ satisfying $0 < \epsilon < \frac{D\sqrt{m_v}}{CkB_2}$. Denoting $\tilde{\phi}'_l$ as the corresponding solution returned by (P), we then have the following results.*

1. *(**General bound**) For any $\gamma \geq 0$:*

$$\| \tilde{\phi}'_l - \phi_l' \|_{L^\infty[-1,1]} \leq \left[ \frac{59(1+\gamma)}{3} \right] \frac{C\epsilon kB_2}{\sqrt{m_v}} + \frac{29}{64m_x^4} \| \phi_l^{(5)} \|_{L^\infty[-1,1]} . \tag{D.4}$$

2. *(**Large step size** $\epsilon$) For any $\gamma > 1$ if $\epsilon$ additionally satisfies*

$$\epsilon > \frac{28\lambda_1}{CkB_2(\gamma-1)} \left( \| \phi_l''' \|_{L^\infty[-1,1]} + \frac{1}{12m_x^2} \| \phi_l^{(5)} \|_{L^\infty[-1,1]} \right) \tag{D.5}$$

*where $\lambda_1 = \Theta(\sqrt{m_x m_v})$ we then have that $\tilde{\phi}'_l$ is a line and:*

$$\| \tilde{\phi}'_l - \phi_l' \|_{L^\infty[-1,1]} \leq \left[ (\gamma+1) + \frac{\gamma-1}{3\lambda} \right] \frac{C\epsilon kB_2}{\sqrt{m_v}} + \frac{29}{64m_x^4} \| \phi_l^{(5)} \|_{L^\infty[-1,1]} . \tag{D.6}$$

The proof simply involves replacing: $g$ with $\phi_l'$, $n+1$ with $2m_x+1$, $h$ with $1/m_x$ and $\tau$ with $\frac{C\epsilon kB_2}{2\sqrt{m_v}}$. As the perturbation $\tau$ is directly proportional to the step size $\epsilon$, hence the large perturbation scenario in Theorem 6 translates to a lower bound on $\epsilon$.

Note that Lemma 1 together with Corollary 3 almost completes the proof of Theorem 8. What remains to be shown is that the bound on $\| \phi_{est,l} - \phi_l \|_{L^\infty[-1,1]}$ is at most a constant times the bound on $\| \tilde{\phi}'_l - \phi_l' \|_{L^\infty[-1,1]}$ for each $l \in S$. This is made precise in the following lemma.

**Lemma 7.** *Let $\alpha$ denote the error bounds of (D.4) or (D.6). We then have for all $l \in S$ that:*

$$\| \phi_{est,l} - \phi_l \|_{L^\infty[-1,1]} \leq 3\alpha$$

*with $\phi_{est,l}$ as defined in (4.6).*

*Proof.* We have that $\tilde{\phi}'_l(x) = \phi'_l(x) + p(x)$ for $x \in [-1, 1]$ with $\| p \|_{L^{\infty}[-1,1]} \leq \alpha$. This then gives us for any $x \in [-1, 1]$:

$$\int_{-1}^{x} \tilde{\phi}'_l(y)dy = \int_{-1}^{x} \phi'_l(y)dy + \int_{-1}^{x} p(y)dy, \tag{D.7}$$

$$\Rightarrow \psi_l(x) - \psi_l(-1) = \phi_l(x) - \phi_l(-1) + \int_{-1}^{x} p(y)dy, \tag{D.8}$$

$$\Rightarrow \int_{-1}^{1} \psi_l(x)dx - \int_{-1}^{1} \psi_l(-1)dx = \underbrace{\int_{-1}^{1} \phi_l(x)dx}_{0} - \int_{-1}^{1} \phi_l(-1)dx + \int_{-1}^{1}\int_{-1}^{x} p(y)dydx, \tag{D.9}$$

$$\Rightarrow \int_{-1}^{1} \psi_l(x)dx = 2(\psi_l(-1) - \phi_l(-1)) + \int_{-1}^{1}\int_{-1}^{x} p(y)dydx. \tag{D.10}$$

By making use of (D.10) we thus have for any $x \in [-1, 1]$:

$$|\phi_{\text{est},l}(x) - \phi_l(x)| = \left| \psi_l(x) - \frac{1}{2}\int_{-1}^{1} \psi_l(x)dx - \phi_l(x) \right|, \tag{D.11}$$

$$= \left| \psi_l(x) - \psi_l(-1) - (\phi_l(x) - \phi_l(-1)) - \frac{1}{2}\int_{-1}^{1}\int_{-1}^{x} p(y)dydx \right|, \tag{D.12}$$

$$\leq \left| \int_{-1}^{x} (\tilde{\phi}'(y) - \phi'_l(y))dy \right| + \frac{1}{2}\int_{-1}^{1}\int_{-1}^{x} |p(y)|\, dydx, \tag{D.13}$$

$$\leq \int_{-1}^{x} |p(y)|\, dy + \frac{1}{2}\int_{-1}^{1}\int_{-1}^{x} |p(y)|\, dydx, \tag{D.14}$$

$$\leq 2\alpha + \frac{\alpha}{2}\int_{-1}^{1} (1 + x)dx = 3\alpha. \tag{D.15}$$

$\square$

Hence Lemma 7, Corollary 3 and Lemma 1 together complete the proof of Theorem 8.

# E Proof of Theorem 3

*Proof.* Recall that the noisy linear system now has the form: $\mathbf{y}_i = \mathbf{V}\mathbf{x}_i + \mathbf{n}_i + \mathbf{z}_i$ where $z_{i,j} = (z'_{i,j} - z'_i)/\epsilon$ is the external noise component and $n_{i,j} = \frac{\epsilon}{2}\sum_{l \in S} v_{j,l}\phi''_l(\zeta^{(l)}_{i,j})v_{j,l}$ is the Taylor's remainder term for $i = -m_x, \ldots, m_x$ and $j = 1, \ldots, m_v$. We saw in the proof of Corollary 2 that $|n_{i,j}| \leq \frac{\epsilon k B_2}{2m_v}$. This gives us:

$$\| \mathbf{z}_i + \mathbf{n}_i \|_{\infty} \leq \left( \frac{2\kappa}{\epsilon} + \frac{\epsilon k B_2}{2m_v} \right) \text{ and } \| \mathbf{z}_i + \mathbf{n}_i \|_2 \leq \sqrt{m_v}\left( \frac{2\kappa}{\epsilon} + \frac{\epsilon k B_2}{2m_v} \right). \tag{E.1}$$

Recall that $\widehat{\mathbf{x}}_i$ is the solution returned by $\ell_1$ minimization (B.1). By proceeding as in the proof of Corollary 2 we obtain for some constant $C > 0$ that

$$\| \widehat{\mathbf{x}}_i - \mathbf{x}_i \|_2 \leq \frac{2C\sqrt{m_v}\kappa}{\epsilon} + \frac{C\epsilon k B_2}{2\sqrt{m_v}} \tag{E.2}$$

holds with high probability for $i = m_x, \ldots, m_x$. As discussed in the proof of Lemma 4 in Section B, if we denote the R.H.S of (E.2) by $\tau$ then the thresholding procedure (B.6) with threshold $\tau$ recovers $S$ exactly if $\tau < D/2$ holds. This is guaranteed if:

$$\frac{2C\sqrt{m_v}\kappa}{\epsilon} + \frac{C\epsilon k B_2}{2\sqrt{m_v}} < D/2 \tag{E.3}$$

$$\frac{C k B_2}{2\sqrt{m_v}}\epsilon^2 - \frac{D}{2}\epsilon + 2C\kappa\sqrt{m_v} < 0. \tag{E.4}$$

Clearly (E.4) has a solution if $\frac{D^2}{4} - 4C^2\kappa k B_2 > 0$ holds. In this case we then obtain the stated condition on $\epsilon$ as the solution range of (E.4). The rest follows in a straightforward manner as for the noiseless setting. $\square$

## F Proof of Theorem 4

*Proof.* After resampling and averaging we have that:

$$\mathbf{z}_i = \left[ \frac{(z'_{i,1} - z'_i)}{\epsilon} \; \frac{(z'_{i,2} - z'_i)}{\epsilon} \; \cdots \; \frac{(z'_{i,m_v} - z'_i)}{\epsilon} \right] ; i = -m_x, \dots, m_x \tag{F.1}$$

where $z'_{i,j}, z'_i \sim \mathcal{N}(0, \frac{\sigma^2}{N})$ are i.i.d $\forall i, j$. We would like to ensure that $\left| z'_{i,j} - z'_i \right| < 2\kappa$ holds for any $\kappa > 0$ and $\forall i, j$ with high probability. Indeed we then obtain a bounded noise model and can simply use the analysis for the setting of arbitrary bounded noise from Section E.

Note that $z'_{i,j} - z'_i \sim \mathcal{N}(0, \frac{2\sigma^2}{N})$. It can be shown that for any $X \sim \mathcal{N}(0, 1)$ we have:

$$\mathbb{P}(|X| > t) \leq \frac{2e^{-t^2/2}}{t}, \quad \forall t > 0. \tag{F.2}$$

Since $z'_{i,j} - z'_i = \sigma \sqrt{\frac{2}{N}} X$ therefore for any $\kappa > 0$ we have that:

$$\mathbb{P}(\left| z'_{i,j} - z'_i \right| > 2\kappa) = \mathbb{P}\left( |X| > \frac{2\kappa}{\sigma} \sqrt{\frac{N}{2}} \right) \tag{F.3}$$

$$\leq \frac{\sigma}{\kappa} \sqrt{\frac{2}{N}} \exp\left( -\frac{\kappa^2 N}{\sigma^2} \right) \tag{F.4}$$

$$\leq \frac{\sqrt{2}\sigma}{\kappa} \exp\left( -\frac{\kappa^2 N}{\sigma^2} \right). \tag{F.5}$$

By taking a union bound over all $i = -m_x, \dots, m_x$ and $j = 1, \dots, m_v$ we have that

$$\mathbb{P}\left( \left| z'_{i,j} - z'_i \right| > 2\kappa : \forall i, j \right) \leq |\mathcal{X}| \, |\mathcal{V}| \frac{\sqrt{2}\sigma}{\kappa} \exp\left( -\frac{\kappa^2 N}{\sigma^2} \right). \tag{F.6}$$

For any $0 < p < 1$ we then have that $\mathbb{P}\left( \left| z'_{i,j} - z'_i \right| > 2\kappa : \forall i, j \right) < p$ if $N$ satisfies:

$$N > \frac{\sigma^2}{\kappa^2} \log\left( \frac{\sqrt{2}\sigma}{\kappa p} |\mathcal{X}| \, |\mathcal{V}| \right). \tag{F.7}$$

This completes the proof. $\qquad \square$