[Reviews · NeurIPS 2014]

Submitted by Assigned_Reviewer_2

This paper considers the problem of estimating the components of a high-dimensional function f(x), where x is a vector of dimension d, which follows a sparse additive model, i.e. f(x) can be written as the sum of a small number of functions f_i(x_i). The authors propose sampling this function at regular points on a certain straight line and additionally on random vertices of a small hypercube surrounding these points. A convex program is then used to obtain an approximation of the gradient of f, exploiting the assumption that it is sparse. 4th order splines are employed to estimate the components of f from this approximation. The authors provide theoretical analysis showing that the method is guaranteed to detect the correct components of f in the absence of noise and also that the procedure is robust to noise.

The main contribution of this paper is an innovative approach to perform robust estimation of gradients in high dimensions through randomized sampling. Although the theoretical analysis follows from standard compressed-sensing results, the idea is clever and the connection to regression with sparse additive models is intriguing. The second step of the algorithm, where the gradient estimates are used to estimate the individual components is less innovative. The authors also provide some useful analysis of the stability of the algorithm, which provides insight into the relationship between the grid size, the curvature of the components and robustness to noise (unfortunately, no numerical simulations are provided to support the theoretical conclusions).

The following are the weakest aspects of the paper:

-The authors do not exploit the group-sparse structure that is implicit in their model. They propose solving a large number of sparse regression problems that have the same sparsity pattern. This suggests two ways of improving the algorithm in terms of sampling complexity and robustness to noise:

1. Estimate the sparsity pattern from a few samples. Then just estimate the derivative of the nonzero components by using a finite-difference approximation in their direction.

2. Employ a regularizer that promotes group sparsity while estimating the gradients, e.g. an l1-l2 norm penalty.

-The data, i.e. the samples of the function of f, are only used to approximate the gradients of the components f_i. They are not taken into account when the component functions f_i are constructed from the gradient estimate. It would probably make sense to ensure that the sum of the estimated components at each sample is consistent with the data up to the noise level.

-The authors do not motivate sufficiently why the model would be of interest in practical applications. A concise example of an application would go a long way to clarify this.

-The numerical simulations are extremely limited. In particular, no attempt is made to illustrate the limits of the method or how the method’s performance depends on the different parameters.

Quality:

The paper makes an interesting algorithmic contribution and is technically sound, although there are some caveats that I have mentioned above. The numerical simulations are insufficient.

Clarity:

The paper is written clearly and is easy to read.

Originality:

The idea of approximating a sparse gradient in high dimensions by sampling a function in random directions within a neighborhood is new to the best of my knowledge. However, recovering a function from its samples by penalizing the norm of its second derivative is not as innovative.

Significance:

This paper is significant from a theoretical point of view in that it provides a method for estimating gradients of a function in a neighbourhood by randomized sampling and convex programming. I am hesitant to state that this is very significant because the authors do not describe any relevant applications and because of the lack of numerical experiments.

Other comments:

The recovery of the components from their gradients must necessarily depend on the second-order behaviour of the components with respect to the chosen grid. A bound on the second derivative does appear in the error bounds, but the authors do not elaborate on its significance.

After the author's rebuttal my views on the paper have not changed. Just to be clear, according to the authors' description of the model the sparsity pattern of the sparse recovery problems they solve is fixed and determined by the set S.
Summary: This paper presents an algorithm to estimate a function following a sparse additive model by first estimating its gradient via compressed sensing and then reconstructing the function from the estimated gradient. The algorithm does not fully exploit the structure in the model and the numerical simulations are very limited, but the method is interesting and novel.

Submitted by Assigned_Reviewer_21

This paper proposed an efficient sampling algorithm for leanring sparse additive model. The paper is well-written and makes improvement over many previous algorithms. The proposed algorithm first recognized that the gradient of the function can be recovered via a CS stage and a further convex QP is utilized to facilitate the final estimation thereafter. I have the following comments for the authors to improve the draft.

1. In the programming (P), what are L'' and L'''? It seems they are not previously defined.

2. Although the theoretical analysis is nice, it would be helpful to present or illustrate some real applications of the theory. The numerical example presented is synthetic and relatvely simple.
Summary: The draft proposed a nice sampling scheme and stronger theoretical bounds have been presented. Some real numerical examples would be helpful to improve the significance of the paper.

Submitted by Assigned_Reviewer_35

This paper considers the problem of learning sparse additive models assuming the target function's values can be calculated at the points of choice. A major improvement the paper achieves over [5,7,8] is the reduced sampling complexity - at a sparsity level of k, the proposed method requires querying the function at only O(k) locations while previous methods require O(c^k) query locations. In addition, the paper provides error bounds for recovering individual basis functions while the error bound provided in many previous works are concerned with recovering the target function (although the methods in [13,14] seem also to estimate individual basis functions).

Technically, the paper takes a multi-step approach: (i) it recovers the subset of active basis functions that actually constitute the target function, and the discrete samples of the differentials of the active basis functions; (ii) it finds a B-spline approximation to each active differential basis functions by solving a quadratic program (QP), (iii) finally the active basis functions are estimated by piece-wise integrals of the differential functions obtained in (ii). The paper provides an upper bound of the estimation error, obtained by analyzing each stage of estimation and integrating the analyses.

The compressive-sensing (CS) formulation employed in stage (i) is interesting, which, along with the QP employed in (ii) to recover each differential basis function from its discrete samples, seems to constitute the main novelty of this paper and makes it distinguished from many existing methods.

Experiments are the weakest part of this paper. No comparison is made to existing methods, and so how these experiments have supported the theoretical findings is quite unclear. One suggestion is to design an experiment to compare the proposed method to those in [5,7,8], and the performance can be measured by the number of function samples (query points) versus the function estimation error. Such an experiment can serve the purpose of demonstrating the reduced sampling complexity achieved by the proposed method.

Summary: The paper describes a new method for sampling a unknown target function which is assumed to follow a sparse additive model. The CS and QP formulations are interesting and apparently novel, and the theoretic analysis seems impressive. The experiments are expected to be enhanced to support the conclusions from the theoretic analysis.
Author Feedback
Author rebuttal: We thank the reviewers for their feedback/suggestions.

We would like to clarify our main contributions, since this has been stated slightly incorrectly in the review.
Estimating sparse gradients via compressed sensing, has been considered previously by Fornasier et al. [8].
However, as stated in the "Related works" section, they consider a substantially different class of functions from us. Hence their sampling scheme also differs considerably from ours, and is not tailored for learning SPAMS.

Our main contribution is related specifically to providing uniform approximation guarantees for each active component
in the SPAM. Furthermore, the QP stage of the algorithm is significant in our opinion, as: 1) it searches over the space of ''not-a-knot" cubic splines and, 2) handles arbitrary but bounded noise.
Such a formulation has not been considered previously, to the best of our knowledge.

(A) Assigned_reviwer_2
-- The reviewer suggests two ways of improving the algorithm, however we do not completely agree with the suggestions.
Firstly, in case we first estimate the sparsity pattern and then estimate the derivatives of the non-zero components (using finite
differences) then this sampling would be less efficient then ours. Estimating the sparsity pattern would need O(k logd) samples, however
the finite difference step would additionally need \Omega(k) samples.

Secondly, it is not clear why there is an implicit group sparse structure in our model, as mentioned by the reviewer. There is no specfic
structured sparsity pattern in our model for us to exploit - we are simply working with arbitrary k sparse vectors.
So there does not appear to be any motivation to employ a regularizer that promotes group sparsity, while estimating the gradients.

-- Ensure sum of estimated components consistent with data : It seems unlikely that this would help in better estimation of the individual components, since we would not know which component(s) is the cause for the overall error. However, it might make sense in the estimation of the function f.

-- Practical applications: One precise application would be to predict gene expression levels based on information on the DNA sequence. Experimental results in this regard are given by Meier et al. [14]. We will elaborate on such applications in the revised version.

-- Significance of second derivative term in bounds: Thank you for the suggestion, we will elaborate on this in the future revisions.

(B) Assigned_reviewer_21
-- L'' and L''' refer to second and third order derivatives respectively. This notation is explained at the beginning of Section 2.

(C) Assigned_reviewer_35
-- Yes, the methods in [13,14] do recover the individual basis functions, however no error bounds are provided for their estimation.
On the other hand, we provide uniform approximation guarantees for each basis function.

GENERAL COMMENT: The numerical simulations that we provide, are only for proof of concept. Our aim was to focus more on the theoretical
aspect of the problem - extensive experimental results will be provided in the expanded version of the paper.